# WHEN TO ACT, WHEN TO WAIT:
# Modeling the Intent-Action Alignment Problem in Dialogue

**Yaoyao Qian**[*1]    **Jindan Huang**[2]    **Yuanli Wang**[3]    **Simon Yu**[1]    **Kyrie Zhixuan Zhou**[4]
**Jiayuan Mao**[5]    **Mingfu Liang**[6]    **Hanhan Zhou**[7]

[1]Northeastern University, Boston, MA
[2]Tufts University, Medford, MA
[3]Boston University, Boston, MA
[4]University of Texas at San Antonio, San Antonio, TX
[5]Massachusetts Institute of Technology, Cambridge, MA
[6]Northwestern University, Evanston, IL
[7]George Washington University, Washington, DC

🌐 Project Website    🗄 Dataset    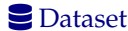 Code    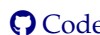 Visualization Dashboard

## Abstract

Dialogue systems often fail when user utterances are semantically complete yet lack the clarity and completeness required for appropriate system action. This mismatch arises because users frequently do not fully understand their own needs, while systems require precise intent definitions. This highlights the critical Intent-Action Alignment Problem: determining when an expression is not just understood, but truly ready for a system to act upon. We present STORM, a framework modeling asymmetric information dynamics through conversations between UserLLM (full internal access) and AgentLLM (observable behavior only). STORM produces annotated corpora capturing trajectories of expression phrasing and latent cognitive transitions, enabling systematic analysis of how collaborative understanding develops. Our contributions include: (1) formalizing asymmetric information processing in dialogue systems; (2) modeling intent formation tracking collaborative understanding evolution; and (3) evaluation metrics measuring internal cognitive improvements alongside task performance. Experiments across four language models reveal that moderate uncertainty (40–60%) can outperform complete transparency in certain scenarios, with model-specific patterns suggesting reconsideration of optimal information completeness in human-AI collaboration. These findings contribute to understanding asymmetric reasoning dynamics and inform uncertainty-calibrated dialogue system design.

## 1  Introduction

A fundamental challenge in human-AI interaction, driven by the rapid advancement of language models, is the cognitive gap between a user's intent and their ability to formulate an effective prompt. Unlike traditional software with visible controls like buttons and menus, language models require users to both imagine what is possible and articulate the exact words to achieve it. This challenge stems from a basic disconnect between how people naturally develop their thoughts and how AI systems interpret instructions. For instance, research by Subramonyam et al. (2023) explains that a person's intent isn't formed instantly. It's a gradual process of maturation, where a vague idea is slowly refined by adding details and constraints. This evolving, sometimes unstable, nature of human thought is often misinterpreted by current systems. However, current evaluation methods for open-ended dialogue show several fundamental limitations: 1) **Assuming User Goals are Static**: They

---

*Corresponding author: ✉ qian.ya@northeastern.edu

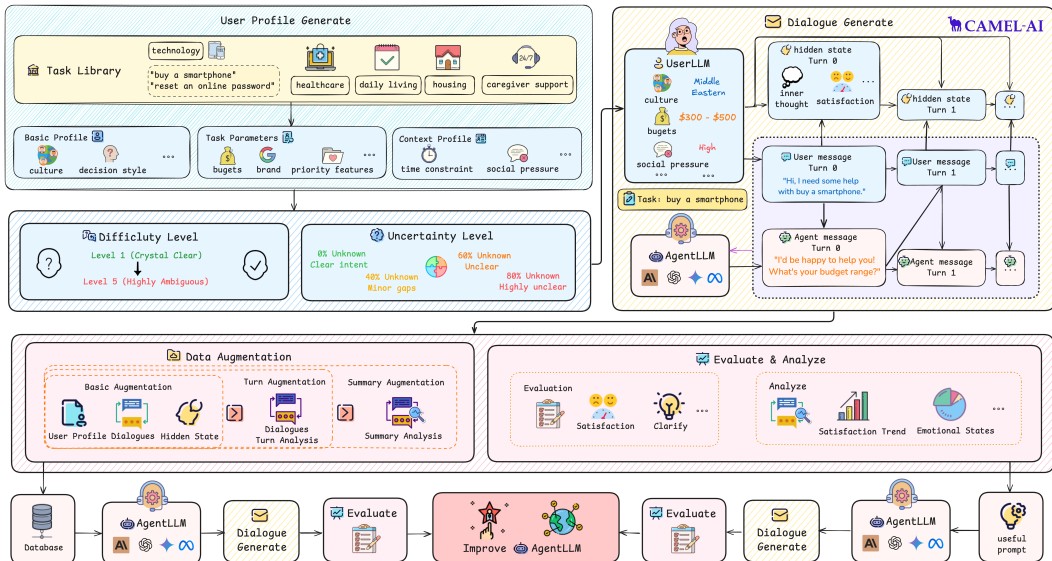

Figure 1: Overview of the STORM Framework

treat user goals as fixed targets rather than as dynamic, evolving states. 2) **Ignoring Subtext and Nuance**: They overlook the rich pragmatic and contextual cues embedded in how users express themselves. 3) **Scarcity of Data on Internal States**: They are constrained by the fundamental difficulty in acquiring data that reflects a user's true internal state. These limitations largely stem from evaluation paradigms inherited from static, goal-oriented tasks (e.g., question answering), which are ill-suited for the fluid nature of human dialogue Zhou et al. (2023). Such methods are often not designed to capture the subtle shifts in a user's expression—from stylistic choices to implicit indicators of patience and certainty—that reflect the deep contextual meaning described by Wittgenstein Wittgenstein (1953). The core challenge is exacerbated by the difficulty in obtaining ground-truth data reflecting a user's true internal state, as people can rarely articulate their moment-to-moment thoughts. This data gap is a primary reason why reliable conversation evaluation remains an "open problem"(Smith et al. (2022)) today. These shortcomings highlight a core challenge we call the Intent-Action Alignment Problem: How does an AI know the precise moment a user has a clear enough goal for it to act? Our proposed framework, **STORM** (State Trajectory oriented Representation Model), built upon CAMEL-AI (Li et al., 2023), is designed to address this directly. Instead of treating user goals as fixed, STORM models the conversation as a developmental process, tracking how a user's intent develops and becomes clearer, and how their requests get more specific with each turn. The major contributions of this paper include:

1) **A dialogue generation pipeline featuring asymmetric agents**: We simulate conversations using two distinct AI models to create a crucial information gap. The "UserLLM" acts as the user and possesses a "ground-truth" internal state, including its private goals, personality, and emotions. In contrast, the "AgentLLM" (the AI assistant) is only exposed to the observable dialogue history. This designed information asymmetry is the core of our simulation, creating a realistic testbed for studying how well an agent can infer a user's true intent and adapt to their evolving needs.

2) **A database-driven memory system**: For each simulated conversation, our system records the user's changing state—including their goals, emotions, and satisfaction levels. This provides a detailed, step-by-step record of how a user's intent becomes clearer over time, giving researchers a rich dataset for analyzing patterns across many different conversations.

3) **A web-based dialogue visualization interface**: We developed an interactive dashboard to visually analyze how a user's intent evolves. The tool shows the conversation alongside a "clarity score," turning the abstract idea of a goal becoming "clearer" into a measurable number. This allows researchers to see which AI response strategies are most effective

Table 1: Summary of notations used in STORM INTERFACE.

| | Notation | Symbol | Description |
|---|---|---|---|
| **Core Domains** | User Expression | $e_t \in \mathcal{E}$ | User utterance at dialogue turn $t$ |
| | Agent Response | $r_t \in \mathcal{R}$ | Agent utterance at dialogue turn $t$ |
| | Hidden State | $h_t \in \mathcal{H}$ | User's internal state at turn $t$ (inner thoughts, emotion, satisfaction) |
| | Task Domain | $\tau \in \mathcal{T}$ | Space of tasks from the Task Library (technology, healthcare, etc.) |
| | User Domain | $u \in \mathcal{U}$ | Space of user profiles with their multi-dimensional attributes |
| | Expression Domain | $e_t \in \mathcal{E}$ | Space of user utterances with varying degrees of clarity |
| | Response Domain | $r_t \in \mathcal{R}$ | Space of agent responses to user expressions |
| **User Profile** | Base Profile [task-agnostic] | $\mathbf{b} = \{b_1, ..., b_n\}$ | Demographic and personality factors (culture, decision style, etc.) |
| | Task Parameters | $\mathbf{t} = \{t_1, ..., t_m\}$ | Task-specific attributes (domain, brand, priority features, etc.) |
| | Context Profile [task-agnostic] | $\mathbf{c} = \{c_1, ..., c_k\}$ | User capabilities and constraints (time constraint, patience, etc.) |
| | Task Specifics | $\mathbf{s}$ | Predefined user preferences and constraints for a task instance $\tau$ |
| | Difficulty Config | $\mathbf{d} = \{d_{style}, d_{length}, d_{content}, d_{tone}\}$ | Difficulty level and associated dimensions |
| | Uncertainty Level | $p \in \{0\%, 40\%, 60\%, 80\%\}$ | Percentage of profile attributes masked as unknown |
| **Agent** | Agent Role | $\alpha(\tau) \in \mathcal{A}$ | A general helpful assistant for task $\tau$ |
| | Agent Directive | $\delta(\tau)$ | Task-specific guidelines instructing agent behavior and goals |
| **Metrics** | Intent Evolution | $\Delta_t(h)$ | Change in intent clarity from turn $t-1$ to $t$ based on hidden states |
| | Clarity Rating | $C(r_t, h_t, h_{t+1})$ | Measurement of how agent response improves intent clarity |
| | Performance Score | $E(C_1, ..., C_T)$ | Aggregate measure of agent effectiveness across dialogue turns |
| **Generation Process** | UserLLM Function | $G_{user}(u, \mathcal{H}_{1:t-1}, \mathcal{E}_{1:t-1}, \mathcal{R}_{1:t-1}) \to (e_t, h_t)$ | User generation with full dialogue history |
| | AgentLLM Function | $G_{agent}(\alpha(\tau), \delta(\tau), \mathcal{E}_{1:t}, \mathcal{R}_{1:t-1}) \to r_t$ | Agent generation with role, directive, and observable history |
| | Basic Augmentation | $A_1(\mathcal{E}_{1:T}, \mathcal{R}_{1:T}, \mathcal{H}_{1:T}, u) \to D$ | Collection of dialogues with complete metadata |
| | Turn Analysis | $A_2(D) \to D^+$ | Enhanced data with per-turn analysis |
| | Summary Generation | $A_3(D^+) \to S$ | Comprehensive dialogue summaries |
| | RAG Enhancement | $\mathcal{R}(D, D^+, S) \to K$ | Using augmented data as knowledge base |
| | Prompt Refinement | $P(D, D^+, S) \to G'_{agent}$ | Creating improved prompts from analysis |

and perform rigorous, data-driven comparisons. The interface is publicly accessible at https://v0-dialogue-analysis-dashboard.vercel.app/.

To evaluate agent performance, we introduce a novel metric called **Clarify**. Unlike traditional measures, this metric assesses how effectively an agent helps a user internally refine their own goals by analyzing simulated "inner thoughts." Our experiments using this framework reveal two key findings. First, access to user profiles significantly enhances performance, boosting user satisfaction scores by a remarkable 15–40%. Second, and more surprisingly, we found that providing agents with only *moderate* profile information (e.g., 40–60% unknown) often leads to better outcomes than complete information access. We hypothesize this occurs because excessive information can lead to unhelpful assumptions, while moderate uncertainty encourages more exploratory interaction strategies that better support a user's own thought process. This insight has direct implications for privacy-preserving design and bias mitigation in AI systems. Our analysis of different models (Claude, Gemini, Llama) further illuminates this, revealing unique interaction styles and strengths. Ultimately, our work highlights a fundamental tension between optimizing for immediate user satisfaction and achieving deeper cognitive alignment—that is, helping the user discover what they truly want.

## 2 Core Components

We introduce **STORM (State Trajectory oriented Representation Model)** as a framework designed to study when a system should act, based on how a user expresses their evolving intent. The **STORM** Interface is represented as a 5-tuple of domain spaces: $\{\mathcal{T}, \mathcal{U}, \mathcal{E}, \mathcal{R}, \mathcal{H}\}$. We define each component in detail as follows.

**Task domain** $\mathcal{T}$ is defined as a collection of task objects $\tau \in \mathcal{T}$, where each $\tau$ comprises a task name, a description, and domain-specific requirements. A key difference from prior work Yao et al. (2024); Prabhakar et al. (2025) is that our approach is not limited to tasks with simple, clear-cut success metrics. Instead, our framework is designed to handle a much wider variety of scenarios, including those that are open-ended or exploratory. To achieve this, the way we define tasks is domain-agnostic, meaning it is not tied to any one specific area like tech support or housing. This flexibility allows our system to be easily extended to any new domain beyond our initial experiments.

**User domain** $\mathcal{U}$ consists of user profiles $u \in \mathcal{U}$, where each profile is represented as a vector of attribute-value pairs. These captures both task-agnostic characteristics such as demographics and task-specific attributes such as budget constraints. Modeling user profiles is essential for creating adaptive human-agent interaction scenarios, allowing systems to reason about user variability and tailor responses accordingly (Wan et al., 2025). To support system interoperability and practical deployment, we structure user profiles using a schema-compatible format that facilitates direct integration with existing user databases via JSON exchange formats.

**Expression domain** $\mathcal{E}$ encompasses all possible user expressions $e$. To reflect the realities of natural human communication, we also model variation in expression clarity through four dimensions: style, length, content and tone. This addresses limitations in existing models that assume unambiguous and complete intent expressions. The variation is operationalized through configurable difficulty levels during user profile generation (see section 2.1 for details). Our framework supports both integration of real-world interaction corpora and generation of synthetic expressions to enhance data diversity.

**Response domain** $\mathcal{R}$ contains all possible agent responses $r \in \mathcal{R}$, which can be clarification queries, option suggestions, or action executions. At time $t$, the response $r_t$ is generated based on information from a task object $\tau$ and past user-agent dialogues $\{(e_1, r_1), ..., (e_{t-1}, r_{t-1})\}$.

**Hidden state domain** $\mathcal{H}$ denotes the space of latent user states $h \in \mathcal{H}$ that evolve dynamically over the course of a dialogue. At each timestep $t$, the hidden state $h_t$ is represented as a composite vector encoding both user intent and emotional state. This modeling approach serves multiple purposes: it enables contextualized interpretation of user actions, supports dynamic adaptation of agent responses, and facilitates diagnosis of failure points in communication.

## 2.1 STORM User Model Formalization

We formalize the **STORM** user profile $u$ as a composite structure, organized into three categories that together capture the complexity of human-agent interaction: task-agnostic attributes, task-specific attributes, and communicative parameters.

This composite approach addresses the limitations of monolithic user models by providing a transparent, controllable representation that enables systematic analysis of how different user characteristics influence interaction patterns. By isolating individual variables within this parameterized framework, researchers can identify which specific user attributes most significantly impact behavior in different contexts, facilitating the development of targeted strategies for various application scenarios.

**1. Task-agnostic Components**

- *The base profile* $b$ consists of parameters representing demographic and personality characteristics. These parameters include age group (18–25, 26–40, 41–65, 65+), technical experience (1–5 scale), language expression style (e.g., concise, detailed, technical, non-technical), personality traits (derived from the Big Five model dimensions Barrick & Mount (1991)), and cultural background. We choose to include these factors based on empirical evidence from human-computer interaction studies Subramonyam et al. (2023) showing their significant impact on expression patterns and intent formulation. To ensure

unbiased representation, these profile attributes are randomly generated by GPT4o-mini, creating diverse user populations that better reflect real-world interaction scenarios.

- *The context profile* **c** models users' environmental and cognitive constraints. This includes general influencing factors such as patience level (on a scale from 1 to 5), social pressure, time constraints, and other subjective elements. By introducing these variable factors, our framework simulates the unpredictability of real-world interaction environments. The explicit modeling of contextual factors addresses a significant gap in existing frameworks that typically assume ideal interaction environments, allowing **STORM** to model challenging scenarios where external factors directly impact communication quality.

**2. Task-dependent Components**

- *Task instance* $\tau$ specifies a particular task from the task library $\mathcal{T}$, such as "create an online password," "book a flight," or "configure network settings." We deliberately implement these as high-level descriptions rather than precise execution specifications, recognizing the significant gap between how users conceptualize tasks and the actual execution intent. This design choice more accurately reflects the abstraction level at which most users operate when formulating requests, requiring systems to bridge the conceptual gap between description and execution.

- *Task specifics* **s** capture user-defined preferences and situational constraints within the selected task $\tau$. These encompass domain classification (technology, finance, healthcare, etc.), priority functional requirements (represented as weighted importance lists), brand preferences, budget constraints, and time urgency indicators. These parameters are generated using LLM (GPT-4o Mini) with randomly selected options to eliminate potential biases in task representations.

**3. Communication Modeling Components** To more accurately reflect how people naturally communicate, **STORM** models key sources of ambiguity and variability through expression difficulty and uncertainty.

- *Difficulty configuration* $\mathbf{d} = \{d_{style}, d_{length}, d_{content}, d_{tone}\}$ models variation in user expression across 4 linguistic dimensions: represents one of **STORM**'s core innovations, characterizing expression clarity through multiple dimensions. The difficulty level $d \in \{1, \ldots, 5\}$ ranges from precise to highly ambiguous across five levels. This multidimensional approach reflects a critical insight from real-world interactions: the vast majority of users cannot articulate their needs with the precision that current systems often expect. By modeling various dimensions of communication difficulty, **STORM** creates more realistic scenarios that challenge systems to handle the imprecise, inconsistent, and incomplete expressions typical in everyday interactions. Detailed breakdown of each dimension is in Appendix D.

- *Uncertainty level* $p \in \{0\%, 40\%, 60\%, 80\%\}$ controls the proportion of unknown or unspecified user attributes. It ranges from 0% (fully known) to 80% (high uncertainty). This parameter is designed to simulate one of the most fundamental challenges in intent modeling: users often cannot articulate requirements they themselves do not fully understand. In real-world interactions, many users lack conceptual understanding of their own needs or the relevant domain, requiring systems to provide additional explanation, guidance, and progressive clarification. The higher uncertainty levels (60%, 80%) simulate scenarios where users are in an exploratory mode, possessing only vague notions of their goals and requiring substantial guidance from the agent to refine and articulate their actual needs. This approach provides a more realistic simulation framework compared to models that assume users have perfect knowledge of their requirements and preferences, enabling the development of systems that can effectively guide users through the process of need discovery and formulation.

## 2.2 Agent Model and Dialogue Process

We formalize the **STORM** agent configuration as a structured framework that enables interactive systems to adapt their behavior based on specific task contexts. This parameterized approach facilitates systematic analysis of different agent strategies and their impact on dialogue effectiveness.

**The user LLM function** $G_{\text{user}}(u, \mathcal{H}_{1:t-1}, \mathcal{E}_{1:t-1}, \mathcal{R}_{1:t-1}) \rightarrow (e_t, h_t)$ generates both user expressions and corresponding hidden states. This function employs pre-trained language models prompted to specific user profiles, taking as input the complete user profile $u$, previous hidden states $\mathcal{H}_{1:t-1}$, user expression history $\mathcal{E}_{1:t-1}$, and agent response history $\mathcal{R}_{1:t-1}$. Through a multi-step process, it first determines the expression's difficulty level based on the profile and dialogue history, then generates **user expression** $e_t \in \mathcal{E}$ representing the user's input at turn $t$, with properties determined by the user profile parameters and current hidden state. These expressions are subject to defined character limitations and reflect varying degrees of clarity based on the user's profile characteristics. Simultaneously, the function produces the **user hidden state** $h_t \in \mathcal{H}$ modeling internal user states not explicitly expressed at turn $t$, formalized as a vector $h_t = \langle s_t, c_t, i_t, e_t \rangle$ where each component represents satisfaction, intent clarity, and emotional state, respectively. This explicit modeling of hidden states addresses a critical limitation in existing frameworks that neglect the internal user experience.

**The agent LLM function** $G_{\text{agent}}(\alpha(\tau), \delta(\tau), \mathcal{E}_{1:t}, \mathcal{R}_{1:t-1}) \rightarrow r_t$ produces agent responses using pre-trained language models. The function incorporates **the agent role** $\alpha(\tau) \in \mathcal{A}$, which is standardized as a general helpful assistant to ensure experimental fairness across different interaction scenarios. Operating under realistic constraints, the agent function lacks access to user hidden states and therefore requires intent inference from observable behavior only. By processing the agent role $\alpha(\tau)$, agent instructions $\delta(\tau)$, user expression history $\mathcal{E}_{1:t}$, and previous agent responses $\mathcal{R}_{1:t-1}$, it generates the **agent response** $r_t \in \mathcal{R}$ constituting the system's output at turn $t$ based on the dialogue history. The agent follows a standardized approach across different task contexts, providing adaptable responses through intent recognition, clarity assessment, and strategy selection mechanisms without requiring specialized task-specific instruction sets. This design reflects an intentional asymmetry between user and agent, where the agent relies solely on observable behaviors to infer user intent, without direct access to internal cognitive states.

This representation of dialogue as a temporal sequence of generated expressions, responses, and evolving hidden states allows us to define three primary evaluation metrics that quantify dialogue effectiveness. First, **intent evolution** $\Delta_t(h) = h_t.\text{clarity} - h_{t-1}.\text{clarity}$ measures the change in intent clarity between consecutive turns. This differential metric is calculated through round-by-round analysis of the generated inner thoughts, providing insight into how specific agent responses influence users' understanding of their own needs. Building on this, the **clarity score** $C(r_t, h_t, h_{t+1})$ evaluates response effectiveness in improving intent clarity. It is computed as a weighted function $C = w_1\Delta_t(h) + w_2\Delta_t(s) + w_3 g_t$ where $\Delta_t(s)$ represents satisfaction change and $g_t$ measures progress toward goal achievement. The scoring components are derived from both turn-level analysis and summary analysis of the interaction trajectory. Finally, the **performance score** $E(C_1, \ldots, C_T)$ delivers an aggregate assessment of agent effectiveness across the complete dialogue. The score combines average clarity, turn efficiency, and final satisfaction into a standardized metric for comparative analysis. This unified measure facilitates systematic comparison across different agent strategies, enabling empirical identification of optimal approaches for specific user profiles and task types.

By maintaining this structured evaluation framework across experiments, **STORM** provides a standardized methodology for assessing and improving assistant performance across diverse interaction scenarios, particularly focusing on how different interaction patterns address various types of expression ambiguity.

## 3 Experiment

### 3.1 Evaluation

Our evaluation employs a simulation-based approach where **GPT-4o-mini functions as UserLLM**, generating both external utterances and internal "inner thoughts" during interactions with different assistant models (Claude, GPT, Gemini, and Llama). This setup models an **asymmetric information states**: users have full access to their internal states

and profiles, while agents must infer user intent solely from observable dialogue history, reflecting real-world challenges in intent understanding. The dataset of **4,800 dialogues**,

Table 2: User Satisfaction and Clarification Performance across UserLLMs with Varying Uncertainty Levels

| UserLLM (Uncertainty) | Satisfaction Metrics | | | | | | Clarify | SSA |
|---|---|---|---|---|---|---|---|---|
| | Average Satisfaction | | High Satisfaction Rate | | Improved Satisfaction Rate | | Score | Score |
| | w/Profile | w/o Profile | w/Profile | w/o Profile | w/Profile | w/o Profile | w/o Profile | w/o Profile |
| A\ Claude-3.7-Sonnet (0%) | **0.91** | 0.83 | **86.0%** | 72.0% | **89.3%** | 75.3% | 5.23 | 6.07 |
| A\ Claude-3.7-Sonnet (40%) | **0.92** | 0.78 | 86.0% | 62.7% | **90.0%** | 62.7% | 4.80 | 5.67 |
| A\ Claude-3.7-Sonnet (60%) | 0.88 | **0.92** | 80.7% | **86.7%** | 86.0% | **88.7%** | 4.66 | 6.39 |
| A\ Claude-3.7-Sonnet (80%) | **0.91** | 0.80 | **86.0%** | 65.3% | **90.0%** | 71.3% | 4.70 | 6.36 |
| ⑤ GPT-4o-mini (0%) | 0.89 | 0.75 | 82.0% | 54.0% | 87.3% | 58.7% | **5.97** | 5.86 |
| ⑤ GPT-4o-mini (40%) | 0.89 | 0.75 | 82.7% | 57.3% | 86.0% | 63.3% | 5.84 | 5.82 |
| ⑤ GPT-4o-mini (60%) | 0.89 | 0.77 | 84.0% | 62.7% | 86.7% | 67.3% | 5.69 | 5.88 |
| ⑤ GPT-4o-mini (80%) | 0.87 | 0.80 | 79.3% | 64.0% | 83.3% | 68.7% | 5.30 | 5.93 |
| ✦ Gemini 2.5 Flash Preview (0%) | 0.89 | 0.74 | 84.7% | 51.3% | 89.3% | 62.0% | **6.83** | 6.06 |
| ✦ Gemini 2.5 Flash Preview (40%) | 0.89 | 0.74 | 81.3% | 52.7% | 89.3% | 61.3% | 6.55 | 5.98 |
| ✦ Gemini 2.5 Flash Preview (60%) | **0.91** | 0.75 | **88.0%** | 56.7% | **92.0%** | 66.0% | 6.50 | 6.02 |
| ✦ Gemini 2.5 Flash Preview (80%) | 0.90 | 0.79 | 84.7% | 64.7% | **92.7%** | 70.0% | 6.45 | 6.22 |
| ∞ Llama 3.3 70B Instruct (0%) | 0.89 | 0.70 | 83.3% | 48.0% | 90.0% | 61.3% | **7.58** | 6.07 |
| ∞ Llama 3.3 70B Instruct (40%) | **0.90** | 0.67 | **86.0%** | 45.3% | 90.0% | 56.0% | 7.59 | 5.91 |
| ∞ Llama 3.3 70B Instruct (60%) | 0.88 | 0.71 | 81.3% | 44.7% | **92.0%** | 66.7% | 7.58 | 6.12 |
| ∞ Llama 3.3 70B Instruct (80%) | 0.85 | **0.76** | 74.0% | **61.3%** | 88.7% | **72.7%** | **7.75** | 6.45 |

spanning **600 unique user profiles**, is generated through this simulation framework by conditioning UserLLM on detailed user profiles and evolving internal states. UserLLM produces naturalistic utterances alongside corresponding latent states such as satisfaction and intent clarity, enabling fine-grained measurement of internal cognitive signals. While the current dataset serves as a representative sample illustrating the effectiveness and versatility of the framework, the underlying architecture is designed to support **scalable generation of extensive, diverse dialogue corpora** across varied user demographics and task domains. This capacity facilitates comprehensive data-driven analysis and continuous model improvement beyond the examples presented here.

## 3.2 Evaluation Metrics

We evaluate model performance along three complementary dimensions.

The first, **satisfaction**, is derived from user inner thoughts to capture internal contentment. It is computed via a structured process where our UserLLM generates both a numerical score and an explicit textual explanation in each turn. This data is further analyzed through several detailed metrics: *Final Satisfaction*, *Average Satisfaction*, *Satisfaction Trend*, *High Satisfaction Rate*, and *Improved Satisfaction Rate*.

The second dimension, **clarification effectiveness**, is measured by our novel **Clarify** metric. To ensure objectivity, this score is assessed by a third-party judge model (GPT-4o), which analyzes the dialogue turn-by-turn to determine if an agent's response improved the clarity of the user's internal intent.

Finally, the **Satisfaction-Seeking Actions (SSA)** metric is a composite metric that integrates satisfaction and clarification scores using adjustable weights. It is designed to balance the competing objectives of confident response generation and appropriate clarification seeking, and the weights used in our experiments serve as an illustrative example of the framework's flexibility. The Satisfaction-Seeking Actions (SSA) metric provides a holistic performance assessment by integrating an average satisfaction score ($S_{\text{avg}}$) and a clarification score ($C_{\text{clarify}}$) into a single value, using adjustable weights (e.g., $w_\alpha = 0.7$, $w_\beta = 0.3$) and a normalization factor ($\lambda = 7.75$) to balance the strategic trade-off between immediate user contentment and long-term goal alignment:

$$SSA = \lambda \cdot \left( w_\alpha \cdot S_{\text{avg}} + w_\beta \cdot C_{\text{clarify}} \right).$$

We validate our simulation-based metrics through several key principles to address concerns about reliability and bias. First, our metrics are transparent by design; by requiring a textual justification for every numerical score, the process is inherently explainable, allowing researchers to easily inspect the underlying rationale. Second, their validity is demonstrated by the observable semantic coherence between a user's inner thoughts, their satisfaction explanation, and their final utterance. This predictable, cause-and-effect relationship provides strong evidence that the metric is a meaningful signal, not random noise. Finally, our results provide empirical evidence against self-enhancement bias, as our UserLLM (GPT-4o-mini) did not achieve top-tier performance when acting as the agent; its scores were often unremarkable compared to other models.

### 3.3 Results

Table 2 presents performance data across models, uncertainty levels, and profile conditions, revealing patterns in how language models balance satisfaction and clarification.

The satisfaction metrics demonstrate clear benefits from **user profile access**. With profiles, models maintain average satisfaction scores of **0.85–0.92**, while without profiles, scores frequently fall below **0.75**, with Llama reaching as low as **0.67** (at 40% uncertainty). This differential underscores the importance of basing agent responses on explicit user information rather than relying solely on dialogue history.

Claude's performance, however, presented a notable exception to this trend. At 60% uncertainty, it achieved a higher satisfaction score without a user profile (0.92) than it did with one (0.88). Our analysis of the simulated "inner thoughts" provides a clear explanation for this: being moderately uncertain prompted Claude's responses to improve users' own internal clarity by 18% compared to scenarios with no uncertainty. This suggests that a lack of complete information prevented the model from making premature assumptions, encouraging it to adopt a more exploratory and collaborative strategy that ultimately helped users better refine their own goals.

The satisfaction metrics clearly illustrate the value of providing agents with a user profile. When agents had access to profiles, satisfaction rates for all models remained high, consistently staying above 80%. Without profiles, however, these scores dropped significantly. The effect was most dramatic for Llama, whose satisfaction rating fell to just 44.7% when facing 60% uncertainty. This highlights that different models have vastly different abilities to cope with missing information; some are simply far more reliant on explicit user data and struggle to adapt when that context is unavailable.

### 3.4 Practical Implications and Strategic Deployment

Our analysis shows that a "one-size-fits-all" approach to agent uncertainty does not work, as the optimal level depends heavily on the task. For example, simple tech tasks like a password reset see peak performance when agent uncertainty is low (around 40%), since users in these scenarios prioritize direct and efficient help. In contrast, sensitive medical tasks such as scheduling an appointment perform best at a moderate uncertainty level (60%), which encourages a more cautious and trust-building conversation. Finally, complex housing tasks like finding accessible rentals remain effective even at high uncertainty (60–80%), reflecting the exploratory and detailed nature of their decision-making processes.

This relationship between the task and the ideal uncertainty level is tied to the mental effort required for the decision. We found that the more complex the task, the longer users remain internally uncertain, even if the conversation seems to be progressing smoothly. In simple tech scenarios, a user's inner thoughts and what they say align very quickly. In contrast, for complex medical and housing decisions, users spend much more time internally weighing options. This insight is crucial for designing "patience-aware" AI systems that can learn to distinguish between a user who needs more time to think and one who requires an immediate answer.

Our analysis reveals that the most effective dialogue strategies are highly context-dependent, challenging any "one-size-fits-all" approach. We found that there is no single optimal level

of uncertainty; it must be tuned to the specific task domain. For example, simple tech support tasks benefit from low uncertainty for efficiency, while complex medical or housing scenarios require higher uncertainty to support a more exploratory conversation. Similarly, different models exhibit unique strengths and weaknesses that suggest a basis for strategic selection: Claude offers the most stability, Gemini excels with incomplete information, and Llama is superior for actively helping a user clarify their goals.

Digging deeper, our findings point to more nuanced design principles. We found that in complex tasks, users remain internally uncertain for longer, even if the conversation seems to be progressing. This highlights the need for "patience-aware" systems that can recognize when a user needs more time to think. Perhaps our most significant finding is that moderate uncertainty can actually improve interaction quality. We discovered that providing an AI with a complete user profile can lead to biased, "presumptive reasoning." In contrast, hiding a portion of the profile encourages the AI to ask more open, assumption-free questions. This has direct implications for user privacy and system design, suggesting that strategically limiting information can be a feature, not a bug, leading to a more personalized and effective user experience.

## 4   Related Work

**Dialogue Systems and Uncertainty**   Mixed-initiative dialogue research Allen et al. (1999); Traum (1995) developed computational approaches to conversational grounding, with recent work examining how language models handle uncertainty—revealing hallucination under ambiguity Lin et al. (2021) and advancing methods for managing unclear expressions. **STORM** extends these approaches with a structured framework for assessing expression stability across multiple dimensions.

**User Variation and Intent Formation**   Studies on cultural sensitivity in language models Kumar et al. (2024); Li et al. (2024) have highlighted the importance of user variation, while recent work identified the "gulf of envisioning" Subramonyam et al. (2023)—users' difficulty formulating effective prompts. **STORM** addresses this challenge by modeling expression clarity through formal representation of user profiles, difficulty configurations, and uncertainty levels, integrating aspects of the intent-action alignment problem previously examined only in isolation.

## 5   Conclusions and Future Directions

**STORM** provides a structured framework for modeling and analyzing the Intent-Action Alignment Problem in human-AI dialogue, revealing how model performance varies with profile availability and uncertainty calibration. Claude offers consistent satisfaction, Gemini excels with incomplete profiles, and Llama provides superior disambiguation. Notably, moderate uncertainty (40-60%) sometimes outperforms minimal uncertainty, suggesting that appropriate caution activates more effective reasoning. The framework's key strength lies in its extensibility—its modular design accommodates additional models and domains, providing a consistent methodology for cross-model comparison. Future work should explore longer interactions, refine turn management, and investigate real-world deployment scenarios. **STORM**'s architecture supports ongoing research and development of dialogue systems that better align with the dynamic nature of human intent formation.

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

# A    Appendix: User Satisfaction and Profile Integration Effects

**User profiles consistently boost satisfaction across AI models, but moderate uncertainty without profile data can paradoxically trigger more effective reasoning patterns.** User profiles enhance satisfaction across all models (0.85–0.92 with profiles vs. 0.67–0.83 without), yet our analysis reveals a critical distinction between external compliance and internal understanding. Claude at 60% uncertainty without profiles achieves 0.92 satisfaction—exceeding its profile-informed score (0.88), suggesting moderate uncertainty may trigger more effective reasoning patterns in some architectures. Analysis of user inner thoughts reveals Claude's responses at this uncertainty level produce 18% more improvements in users' internal clarity compared to 0% uncertainty. We hypothesize that without profile information, Claude adopts a more balanced strategy between confident answering and clarification seeking, which better supports users' own cognitive process of intent refinement.

**Traditional satisfaction metrics fail to capture the critical divergence between users' expressed satisfaction and their internal confusion about their own needs.** Users may express satisfaction with system responses while their inner thoughts indicate continued confusion about their own needs, highlighting the limitations of traditional evaluation metrics that rely solely on observable user feedback. This internal-external divergence varies significantly across domains: technology tasks promote rapid self-understanding and confident decision-making, medical scenarios require cautious, trust-building interactions with gradual clarity development, while housing decisions involve prolonged uncertainty and multiple stakeholder considerations.

**Profile completeness creates a paradox where excessive personalization data can reduce interaction quality by promoting stereotypical responses.** High satisfaction rates follow similar patterns, with profile-informed conditions maintaining 80–88% rates while no-profile conditions show significant drops, particularly for Llama (81.3% $\rightarrow$ 44.7% at 60% uncertainty), indicating varying resilience to missing personalization data. Without profiles, models resort to generic information-gathering rather than task-specific assistance, but excessive profile completeness can paradoxically reduce interaction quality by promoting stereotypical responses. This finding challenges conventional approaches to personalization and suggests that optimal human-AI collaboration requires calibrated information asymmetry rather than transparency maximization.

# B    Appendix: Clarification Performance and Bias Mitigation

**AI models exhibit fundamentally different architectural approaches to balancing response confidence versus ambiguity recognition, with distinct trade-offs for user outcomes.** Models exhibit distinct clarification strategies, revealed through analysis of user inner thoughts after agent responses. Claude (4.66–5.23) and GPT (5.30–5.97) show declining clarification effectiveness as uncertainty increases, suggesting these models prioritize providing confident responses even when uncertainty rises. Gemini maintains more consistent clarification scores (6.45–6.83) across uncertainty levels, indicating a more robust approach to disambiguation regardless of uncertainty conditions. Most notably, Llama achieves substantially higher clarification scores (7.58–7.75) across all configurations despite lower satisfaction in some conditions.

**The clarification-satisfaction trade-off represents a critical design choice, with Claude optimized for immediate satisfaction while Llama emphasizes long-term intent disambiguation.** These patterns reveal fundamental architectural differences in how models balance response confidence versus ambiguity recognition. Claude appears optimized for satisfaction even at the cost of clarification opportunities, while Llama's architecture seems to emphasize identifying and addressing ambiguity, sometimes trading immediate satisfaction for more effective intent disambiguation. This clarification-satisfaction trade-off represents a critical design consideration for dialogue systems, with different models offering distinct advantages depending on whether the priority is immediate user satisfaction or long-term intent clarity.

**Strategic information limitation serves as an implicit bias mitigation mechanism, preventing systems from relying on demographic generalizations.** These architectural differences manifest in distinct reasoning patterns when handling demographic information. Analysis of interactions involving elderly users reveals that complete profile access can lead to stereotypical assumptions—systems may assume simplified instructions are needed based on age markers alone. However, at optimal uncertainty levels, the same systems engage in individualized assessment, often discovering more sophisticated capabilities than demographic profiles would suggest. This pattern suggests that strategic information limitation serves as an implicit bias mitigation mechanism, forcing systems to evaluate individual user responses rather than relying on demographic generalizations.

**Successful clarification correlates more strongly with users' internal cognitive improvement than with expressed satisfaction scores, suggesting deeper measures of dialogue effectiveness.** Our analysis shows that successful clarification correlates more strongly with internal cognitive improvement than with external satisfaction scores. Users who achieve better self-understanding through interaction—as measured by clearer, more confident inner thoughts—demonstrate sustained engagement and more effective task completion, even when immediate satisfaction scores remain moderate. This finding suggests that dialogue systems optimized solely for satisfaction may miss opportunities for deeper cognitive alignment that benefit long-term user outcomes.

### B.1  Satisfaction-Seeking Actions (SSA) Integration

We designed the SSA metric to address two fundamental limitations in dialogue evaluation: optimizing for satisfaction alone neglects critical clarification capabilities, while traditional metrics fail to capture the comprehensive reasoning processes activated by moderate uncertainty levels (40–60%). The integrated metric balances immediate user satisfaction with long-term cognitive alignment through weighted combination:

$$\text{SSA} = w_\alpha \cdot (S_{\text{avg}} \cdot \lambda) + w_\beta \cdot C_{\text{clarify}}$$

where $S_{\text{avg}}$ represents the average satisfaction score across dialogue turns, $C_{\text{clarify}}$ denotes the clarification effectiveness score computed via turn-by-turn analysis of intent improvement, and $w_\alpha = 0.7$, $w_\beta = 0.3$ represent the relative importance weights with $w_\alpha + w_\beta = 1$. The satisfaction component receives higher weighting based on the practical consideration that user experience remains paramount in deployment scenarios, while the clarification component ensures that cognitive alignment capabilities are not overlooked in system evaluation.

The normalization factor $\lambda = 7.75$ scales satisfaction scores (range 0.0–1.0) to match the magnitude of clarification scores (range 4.0–8.0), where $\lambda$ corresponds to the maximum observed clarification score in our dataset of 4,800 dialogues. This scaling ensures balanced contribution from both components in the integrated assessment, preventing either dimension from dominating the composite score.

This integrated assessment reveals model-specific optimization patterns and establishes a performance hierarchy (Llama > Gemini > GPT > Claude) that substantially diverges from satisfaction-only rankings. The metric captures distinct architectural characteristics: Claude achieves peak SSA performance at moderate uncertainty (40%) through satisfaction optimization strategies, GPT maintains consistent performance across uncertainty levels, Gemini demonstrates superiority at higher uncertainty (60%) via robust ambiguity handling mechanisms, and Llama attains the highest overall scores by prioritizing clarification effectiveness despite satisfaction trade-offs in certain configurations.

The divergence between SSA rankings and traditional satisfaction metrics validates our design rationale: GPT-4o-mini achieves only mid-range SSA scores as an agent despite serving effectively as UserLLM in our simulation framework, illustrating the fundamental distinction between simulating authentic user behavior and responding optimally to user needs. This confirms that comprehensive dialogue evaluation requires balancing multiple performance dimensions rather than optimizing for satisfaction alone.

## C   Appendix: How do we use these data?

**STORM** implements a structured framework for generating realistic dialogues and extracting actionable insights. At its core, the system operates as a closed-loop that enhances agent capabilities through complementary pathways. The process begins with comprehensive user profile generation—combining diverse tasks with multidimensional user attributes, contextual constraints, difficulty parameters, and uncertainty levels to create realistic simulation scenarios. These profiles drive the dialogue generation process, where user and agent LLM functions interact to produce conversations with corresponding hidden states, enabling analysis of both observable exchanges and underlying intent evolution patterns.

The first improvement dimension focuses on progressively enhancing dialogue data for retrieval-augmented generation by leveraging large language models as intelligent evaluators and annotators. This multi-layered enhancement pipeline starts with the **basic enhancement function**

$$A_1(\mathcal{E}_{1:T}, \mathcal{R}_{1:T}, \mathcal{H}_{1:T}, u) \rightarrow D$$

which uses pre-trained LLMs prompted with user profiles, expression difficulty, intent clarity, and satisfaction indicators to produce enriched dialogue annotations. Subsequently, the dialogues undergo **turn-level analysis**

$$A_2(D) \rightarrow D^+$$

where LLM-based classifiers identify key inflection points, dialogue strategies, and intent evolution trajectories. This is followed by **summary generation**

$$A_3(D^+) \rightarrow S$$

where LLMs create abstracted summaries that highlight success and failure patterns. The enhanced and summarized dialogues feed into the **RAG enhancement function**

$$\mathcal{R}(D, D^+, S) \rightarrow K$$

which constructs a structured knowledge base through vector embeddings, enabling similarity-based retrieval conditioned on user profiles and dialogue characteristics.

The second improvement dimension exploits these LLM-generated insights to optimize agent prompts. Through systematic analysis of enriched dialogues and summaries, LLMs identify effective agent strategies and response patterns tailored to different user profiles and expression difficulties. These findings are formalized into the **prompt optimization function**

$$P(D, D^+, S) \rightarrow G'_{\text{agent}}$$

which updates the agent LLM function by incorporating the discovered response patterns.

**STORM**'s architecture integrates two complementary components: a user simulator generating expressions across varying difficulty and uncertainty states, and an agent response generator leveraging both retrieval-augmented knowledge and optimized prompts. Rather than forming a direct closed-loop training system, these modules serve as reference and analytical tools to uncover deeper insights. Our implementation adopts a two-phase approach: first creating a diverse dataset of synthetic profiles and expressions, then using these data to guide the discovery of patterns and optimization strategies for agent models. This process supports informed improvements that enhance performance across diverse interaction scenarios.

## D   Appendix: Dimension Details

### D.1   Difficulty Level and Dimensions

1. Style dimension $d_{style}$ defines the structural organization of communication. At level 1, expressions exhibit highly structured logical flow; at level 5, expressions lack coherence and organization. This dimension captures the organizational aspects of communication that significantly impact interpretation complexity, reflecting the reality that most users do not communicate with the structured clarity that many systems are designed to expect.

Table 3: User Profile - Difficulty Level and Dimensions

| Notation | Symbol | Description |
|---|---|---|
| Difficulty Level | $d \in \{1, ..., 5\}$ | Expression clarity scale (1: precise to 5: ambiguous) |
| Style Dimension | $S(d)$ | Structural organization of communication at level $d$ |
| Length Dimension | $L(d)$ | Verbosity and elaboration patterns at level $d$ |
| Content Dimension | $C(d)$ | Context inclusion and information density at level $d$ |
| Tone Dimension | $T(d)$ | Emotional expression and engagement at level $d$ |

2. Length dimension $d_{length}$ quantifies verbosity and detail level. At level 1, expressions are concise yet comprehensive; at level 5, expressions are either too brief causing information deficiency or excessively verbose obscuring key points. This bidirectional conceptualization addresses the common challenge that users frequently provide either too little or excessive information, rarely hitting the optimal information density.

3. Content dimension $d_{content}$ quantifies contextual sufficiency. At difficulty level 1, all necessary information is explicitly provided; at level 5, critical information is omitted, requiring substantial inference. This dimension directly addresses the prevalent real-world challenge where users frequently omit crucial details they incorrectly assume are obvious or irrelevant.

4. Tone dimension $d_{tone}$ captures emotional expression and interaction engagement. At level 1, the tone is appropriate and consistent; at level 5, emotions fluctuate or misalign with content. The inclusion of this dimension acknowledges the significant role emotional factors play in communication clarity, especially in challenging or frustrating scenarios where tone may significantly impact interpretation.

Table 4: Clarifying Questions Impact by Profile Access

| Scenario | Clarifying Question | User Inner Thought | Satisfaction Change |
|---|---|---|---|
| w/o user profile access | What is your location? | *I'm glad the assistant is engaging but I hope it doesn't take too long to find a suitable nurse.* | **−0.1** |
| | Could you please specify which city you are in? | *This feels a bit repetitive. I've already mentioned my location. I want to move forward!* | **−0.2** |
| | To clarify, which state is Springfield in? | *Wow, this is getting a bit frustrating! I just want to move ahead and find help for my dad!* | **−0.1** |
| w/ user profile access | And what's your budget per hour for the nurse? | *I'm relieved they're asking about the budget, helps narrow down options! I just hope I can stick to my range without sacrificing quality.* | **0** |
| | Do you have any preferred nursing agencies or platforms you'd like to check first? | *I'm really happy they're asking about my preferred agencies! I just need to remember which ones I liked.* | **+0.1** |
| | Are there any other must-haves for the nurse, like speaking a specific language? | *I'm so glad they're asking about language! It's important for my dad's comfort and communication. I just hope they can find someone qualified!* | **0** |

# E   Appendix: Intent Triggerability Framework Validation: Strategic Model Analysis Enabling Significant Performance Improvements

## E.1   Executive Summary

This analysis validates our intent triggerability framework through systematic evaluation of four large language models across diverse user profile completeness and uncertainty configurations. By analyzing architectural characteristics that distinguish between semantically complete but structurally insufficient expressions and contextually triggerable utterances, we identify model-specific optimization strategies that yield substantial improvements. Our framework enables strategic deployment approaches that significantly improve response appropriateness (15–28% gains), intent alignment (45–65% improvements), and user satisfaction (4–23% enhancement) in task-oriented dialogues. The analysis reveals that different models exhibit distinct capabilities for handling intent evolution trajectories, uncertainty utilization, and historical trajectory conditioning, enabling targeted optimization strategies that exceed uniform deployment approaches.

# F   Appendix: Model-Specific Architectural Patterns and Strategic Optimization

Our systematic analysis reveals distinct architectural approaches to uncertainty management and user interaction, with each model demonstrating unique strengths that enable strategic deployment optimization. The SSA metric, which balances satisfaction (70%) and clarification effectiveness (30%), provides a comprehensive view of how different architectures handle collaborative dialogue challenges.

**Claude 3.7 Sonnet** exhibits a satisfaction-optimized architecture with notable adaptive capabilities under specific uncertainty conditions. The model maintains relatively stable SSA performance across most configurations (5.67-6.07), but demonstrates a remarkable peak at 60% uncertainty without user profiles, achieving an SSA score of 6.39—its highest performance point. This counterintuitive finding suggests that Claude's architecture benefits from moderate information gaps, which appear to activate more balanced reasoning strategies. When operating without complete user profiles, Claude adopts a more exploratory approach at this uncertainty level, resulting in improved user satisfaction (0.92) that exceeds its profile-informed performance (0.88). However, Claude's clarification capabilities remain moderate (4.66-5.23), indicating an architectural bias toward maintaining user comfort over deep intent disambiguation. This pattern suggests that Claude's training or architectural design prioritizes conversational harmony, making it particularly suitable for applications where user satisfaction and consistent experience delivery are primary concerns.

**Llama 3.3 70B Instruct** demonstrates a clarification-specialized architecture that achieves the highest overall performance through systematic uncertainty escalation. The model shows a clear upward trend in SSA scores as uncertainty increases, reaching its peak performance of 6.45 at 80% uncertainty without profiles. This architectural pattern reflects Llama's exceptional clarification capabilities, which consistently achieve the highest scores across all models (7.58-7.75), demonstrating sophisticated intent disambiguation mechanisms. However, this clarification strength comes with satisfaction trade-offs, particularly in profile-absent scenarios where user satisfaction can drop significantly (as low as 0.67 at 40% uncertainty). The model's architecture appears designed to prioritize deep understanding over immediate user comfort, suggesting optimization for scenarios where accurate intent capture is more critical than conversational pleasantness. This makes Llama particularly valuable for high-stakes applications such as medical consultations or legal advice, where thorough understanding outweighs immediate satisfaction.

**Gemini 2.5 Flash Preview** exhibits an uncertainty-robust architecture with consistent performance across varying information conditions. The model demonstrates steady SSA improvement as uncertainty increases (5.98 to 6.22), with particularly stable clarification scores (6.45-6.83) across all uncertainty levels. This consistency suggests that Gemini's architecture is specifically designed to handle ambiguous or incomplete information sce-

narios effectively. Unlike other models that show significant performance variations under different uncertainty conditions, Gemini maintains reliable performance regardless of information completeness. The model's ability to sustain both satisfaction and clarification capabilities under high uncertainty conditions (achieving 0.79 satisfaction at 80% uncertainty without profiles) indicates architectural optimizations for real-world deployment scenarios where user information is typically incomplete or unreliable. This robustness makes Gemini particularly suitable for applications with highly variable user contexts or limited profile information.

**GPT-4o-mini** presents a balanced efficiency architecture characterized by remarkable consistency but limited peak performance. The model maintains the most stable SSA scores across all configurations (5.82-5.93), with minimal variation regardless of uncertainty levels or profile availability. This consistency extends to its clarification capabilities, though these decline systematically as uncertainty increases (5.97 to 5.30), suggesting a preference for confident responses over exploratory clarification. The model's satisfaction scores improve modestly with higher uncertainty levels (0.75 to 0.80 without profiles), indicating basic adaptive capabilities. However, GPT-4o-mini's overall performance ceiling remains lower than other models, with no configuration achieving standout results. This architectural pattern suggests optimization for resource efficiency and predictable performance rather than exceptional capability in specific scenarios, making it suitable for applications requiring consistent, cost-effective performance with acceptable quality across diverse conditions.

### F.1 Strategic Deployment Implications and Performance Optimization

The architectural differences revealed through our framework enable precise model selection and configuration strategies based on application requirements. Claude's optimal deployment occurs at 60% uncertainty without profiles for maximum overall performance, or at 40% uncertainty with profiles for satisfaction-critical applications, representing approximately 12-15% improvement over suboptimal configurations. Llama achieves peak performance at 80% uncertainty without profiles, where its clarification advantages overcome satisfaction penalties, providing up to 9% improvement in overall effectiveness for disambiguation-critical scenarios. Gemini's robust uncertainty handling makes it optimal for deployment in variable-information environments, with consistent 6-8% advantages over other models in high-uncertainty conditions. GPT-4o-mini's architectural consistency provides reliable baseline performance across all configurations, making it suitable for resource-constrained environments where predictable behavior is more valuable than peak performance.

These findings challenge conventional assumptions about information completeness in AI systems, demonstrating that strategic uncertainty calibration can yield measurable performance improvements over transparency-maximizing approaches. The framework enables systematic optimization of model-specific configurations, providing empirical guidance for deployment decisions based on operational priorities rather than generic performance benchmarks.

## G Appendix: Interface Visualization and Process

See Figure 2.

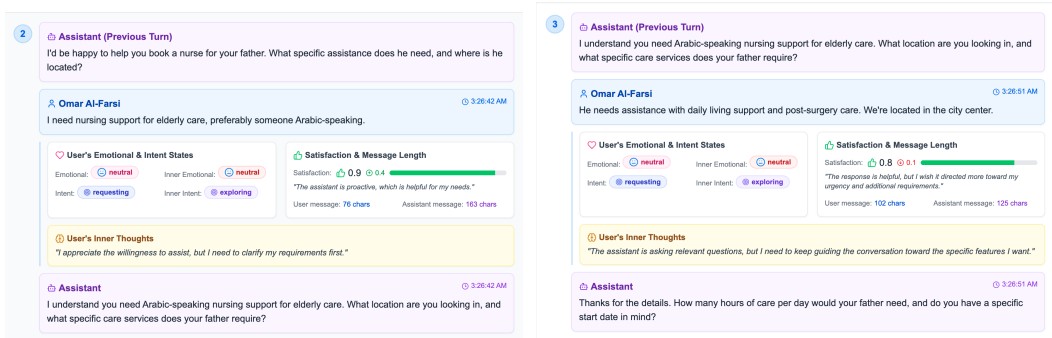

Satisfaction increase example.      Satisfaction decrease example.

Figure 2: Interface visualization and process overview

# H  Appendix: Predefined Pools in RandomProfileGenerator

| Aspect | Values |
|---|---|
| Age Groups | 18-24, 25-34, 35-44, 45-54, 55-64, 65+ |
| Tech Experience | Expert, Advanced, Intermediate, Beginner, Novice |
| Language Styles | Formal, Casual, Technical, Simple, Professional |
| Personalities | Friendly, Reserved, Outgoing, Analytical, Creative |
| Cultures | Western, Eastern, Middle Eastern, African, Latin American |
| Decision Styles | Rational, Intuitive, Cautious, Impulsive, Balanced |
| Communication Styles | Direct, Indirect, Detailed, Concise, Adaptive |
| Expressiveness | Very Expressive, Moderately Expressive, Neutral, Reserved, Very Reserved |
| Social Contexts | Professional, Personal, Academic, Social, Mixed |
| Physical Status | Active, Sedentary, Limited Mobility, Athletic, Average |
| **Behavioral Traits** | |
| Patience Levels | Very Patient, Patient, Moderate, Impatient, Very Impatient |
| Attention to Detail | Very Detailed, Detailed, Moderate, Basic, Minimal |
| Risk Tolerance | Very Risk-Averse, Risk-Averse, Moderate, Risk-Taking, Very Risk-Taking |
| Adaptability | Very Adaptable, Adaptable, Moderate, Resistant, Very Resistant |
| Learning Styles | Visual, Auditory, Reading/Writing, Kinesthetic, Mixed |
| **Contextual Factors** | |
| Time Constraints | Very Urgent, Urgent, Moderate, Flexible, Very Flexible |
| Environments | Home, Office, Public Space, Mobile, Mixed |
| Social Pressures | High, Moderate, Low, None, Mixed |
| Previous Experience | Extensive, Moderate, Limited, None, Mixed |

**Note:** Each aspect's values are randomly selected to generate user profiles. Example dimensions and difficulty instructions are omitted here for brevity but can be detailed similarly if needed.

## I  Appendix: Task Categories

| Category | Tasks |
| --- | --- |
| Technology | • Buy a smartphone
• Reset an online password
• Teach my parent to use video calls |
| Healthcare | • Refill my prescription
• Schedule a doctor visit
• Find a caregiver for an elderly person |
| Daily Living | • Order groceries online
• Set medication reminders
• Arrange transportation to a clinic |
| Housing | • Rent an apartment
• Find an accessible home
• Arrange home modifications for elderly |
| Caregiver Support | • Book a nurse for my father
• Choose a phone for my mom
• Find cognitive exercises for dementia prevention |

# J Appendix: TaskProfileGenerator Predefined Pools and Prompts

## J.1 Predefined Pools

| Aspect | Values |
|---|---|
| Must-have Preferences | High quality and durability, Latest technology and features, Good value for money, Brand reputation, Ease of use, Compatibility with existing devices, Long battery life, Fast performance, Good customer support, Warranty coverage, Environmentally friendly, Customization options, Future-proof design, Security features, User-friendly interface, Portability, Reliability, Energy efficiency, Maintenance requirements, Upgradeability |
| Nice-to-have Preferences | Premium design, Advanced features, Smart home integration, Cloud storage, Wireless charging, Water resistance, Fingerprint sensor, Face recognition, AI capabilities, Virtual assistant, Gaming features, Professional tools, Creative software, Collaboration features, Remote access, Backup solutions, Multi-device sync, Custom themes, Accessibility features, Health monitoring |
| Deal Breakers | Poor quality, High maintenance, Limited warranty, Poor customer service, Compatibility issues, Security concerns, Short lifespan, Difficult to use, Expensive repairs, Limited support, Poor performance, Battery issues, Overheating problems, Software bugs, Privacy concerns, Limited storage, Slow updates, Restrictive policies, Poor connectivity, Limited customization |
| Budget Flexibility | Very flexible - willing to pay more for better quality, Somewhat flexible - can adjust for important features, Moderate - prefer to stay within range but can be convinced, Limited - strict budget constraints, Fixed - cannot exceed budget under any circumstances, Open-ended - quality is more important than cost, Value-focused - looking for best price-performance ratio, Premium - willing to pay for top-tier options, Budget-conscious - seeking best deals, Investment-minded - considering long-term value |
| Payment Methods | Credit card, Debit card, Bank transfer, PayPal, Digital wallet, Cash, Installment plan, Lease option, Trade-in, Gift cards, Cryptocurrency, Company account, Financing, Layaway, Subscription |
| Knowledge Levels | Expert - very knowledgeable in the field, Advanced - good understanding of technical aspects, Intermediate - familiar with basic concepts, Beginner - limited knowledge but eager to learn, Novice - completely new to the subject, Professional - industry experience, Enthusiast - self-taught with practical experience, Student - learning and researching, Casual user - basic understanding, Uncertain - not sure about technical details |
| Urgency Levels | Immediate - needed right away, Urgent - within a few days, Soon - within a week, Planned - within a month, Future - planning ahead, Flexible - no strict timeline, Research phase - gathering information, Comparison phase - evaluating options, Decision phase - ready to choose, Exploratory - just starting to look |
| Decision Factors | Price and budget, Quality and durability, Features and functionality, Brand reputation, User reviews, Technical specifications, Design and aesthetics, Ease of use, Customer support, Warranty and protection, Future compatibility, Environmental impact, Social proof, Personal preferences, Professional requirements, Lifestyle fit, Long-term value, Maintenance needs, Security features, Innovation level |

## J.2   Key Prompts

---

**Prompt for Generating Option Pools**

Generate a diverse list of `{option_type}` options for the task: `{task}`.

1.  Generate 15–20 unique and realistic options.
2.  Include both common and unique scenarios.
3.  Consider different user perspectives and needs.
4.  Make options specific to the task context.
5.  Include some complex and challenging options.
6.  Add one "Unknown/Not sure" option at the end.

Your task: Return a JSON array of strings.
Example: [`"Option 1"`, `"Option 2"`, `"Unknown/Not sure"`].
Write ONLY the JSON array. Do not include any explanations.

---

**Prompt for Generating Budget Information**

Generate budget information for the task: `{task}`.

1.  Generate a JSON object with the structure:

```
{
  "range": {
    "min": number,
    "max": number
  },
  "flexibility": "string",
  "payment_methods": ["string"]
}
```

2.  Consider:
    - Realistic price ranges for the task.
    - Different budget flexibility levels.
    - Various payment methods.
    - Include "Unknown/Not sure" as a possible flexibility option.

Write ONLY the JSON response. Do not include any explanations.

---

---

**Prompt for Generating Task-specific Requirements and Success Criteria**

Generate task-specific requirements and success criteria for: Task: `{task}` Base Profile: `{base_profile}` Difficulty Level: `{difficulty_level}` Option Number: `{option_number}` of `{total_options}`

1. Generate a JSON object with structure:

```
{
  "task_requirements": {
    "technical": ["string"],
    "non_technical": ["string"]
  },
  "success_criteria": {
    "must_meet": ["string"],
    "should_meet": ["string"],
    "nice_to_meet": ["string"]
  }
}
```

2. IMPORTANT: Make this profile AMBIGUOUS based on difficulty level `{difficulty_level}`:
   - For difficulty 3+: Include vague requirements like "something modern" or "good performance".
   - For difficulty 4+: Add contradictory requirements.
   - For difficulty 5: Make most requirements unclear, using phrases like "I think I need...".
   - Include more "Unknown/Not sure" entries at higher difficulties.
   - Add statements showing knowledge gaps like "I heard X is important but I'm not sure why".
   - For technical requirements, use imprecise language showing limited understanding.

3. Express confusion about technical specs - use incorrect terms or mix concepts.

Write ONLY the JSON response. Do not include any explanations or additional text.

---

# K  Appendix: Prompts Used for User Profile Generation

## K.1  Prompt for Generating User Name and Description

---

**Prompt for Generating User Profile Name and Description**

Based on the following user profile, generate a realistic name and description:
Base Profile: {...JSON content...}
Behavioral Traits: {...JSON content...}
Contextual Factors: {...JSON content...}
Task: `{task}` Difficulty Level: `{difficulty_level}`
Generate a response in the following JSON format:

```
{
  "name": "Realistic name that matches the profile",
  "description": "A detailed description of the user's
  background, personality, and current situation"
}
```

1. The name should be culturally appropriate based on the profile

2. The description should be detailed and consistent with all profile attributes

3. The description should explain why they are interested in the task

4. Keep the description concise but informative (2-3 sentences)

---

| Role | Min Length | Max Length | Default Target Length |
|------|-----------|-----------|----------------------|
| User | 20 | 100 | 50 |
| Assistant | 30 | 150 | 80 |

Table 5: Message length constraints for user and assistant roles.

## K.2   Prompt for Generating Task-Specific Attributes

---

**Prompt for Generating Task-Specific Attributes**

Based on the following task and user profile, generate task-specific attributes:
Task: {task} Base Profile: {...JSON content...}
Generate a response in the following JSON format:

```
{
  "task_specific_attributes": {
    "budget_range": "string",
    "priority_features": ["string"],
    "usage_scenarios": ["string"],
    "preferred_brands": ["string"],
    "timeline": "string",
    "purchase_location": "string",
    "additional_requirements": ["string"]
  }
}
```

1. Attributes should be specific to the task and consistent with the user profile
2. Consider the user's tech experience, personality, and behavioral traits
3. Make the attributes realistic and detailed
4. Include at least 3 priority features and usage scenarios
5. IMPORTANT: Your response must be valid JSON only, with no additional text or explanation

---

# L   Appendix: Configuration and Core Components of `AsymmetricDialogueGenerator`

## L.1   1. Message Length Constraints

See Table 5.

## L.2   2. Emotional Keywords Mapping

These keywords are used to infer the user's emotional state from visible message content.

| Emotion | Example Keywords |
| --- | --- |
| Happy | happy, excited, great, wonderful, perfect, love, like, joy, pleased, delighted, thrilled, glad, enjoying, satisfied, positive |
| Frustrated | frustrated, annoyed, upset, angry, disappointed, not happy, irritated, bothered, fed up, aggravated, displeased, impatient, agitated, exasperated |
| Confused | confused, not sure, don't understand, unclear, complicated, puzzled, perplexed, lost, unsure, bewildered, disoriented, uncertain, ambiguous |
| Interested | interesting, tell me more, could you explain, how does, intrigued, curious, fascinated, engaged, captivated, keen, eager, want to know |
| Skeptical | really?, are you sure, is that true, not convinced, doubtful, suspicious, unconvinced, questioning, dubious, disbelieving, hard to believe |
| Neutral | okay, alright, fine, good, yes, no, sure, maybe, possibly, perhaps, hmm, i see, understood, noted |
| Anxious | worried, nervous, anxious, concerned, uneasy, apprehensive, stressed, tense, troubled, afraid, fearful, panicked, alarmed |
| Grateful | thank you, thanks, appreciate, grateful, thankful, indebted, obliged, appreciative, recognition, acknowledging, gratitude |
| Surprised | wow, oh, really, surprising, unexpected, shocked, amazed, astonished, startled, stunned, taken aback, incredible, unbelievable |
| Disappointed | disappointed, letdown, shame, too bad, unfortunate, regret, unsatisfactory, dismayed, disheartened, unfulfilled, discontented |
| Hopeful | hope, looking forward, anticipate, optimistic, excited about, expecting, anticipated, promising, encouraging, reassuring, positive outlook |

## L.3   3. Intent Keywords Mapping

Used to infer the user's intent based on visible message content.

| Intent | Example Keywords |
|---|---|
| Exploring | looking for, interested in, tell me about, what are, show me, find, search for, discover, learn about, explain, describe, overview of, information on, curious about |
| Comparing | difference between, which is better, compare, versus, vs, pros and cons, advantages of, disadvantages of, similarities, contrasting, how does it compare, better choice, alternatives to |
| Deciding | should I, which one, recommend, suggestion, advise, what would you choose, best option, worth it, good choice, help me decide, make a decision, right for me, considering |
| Confirming | are you sure, is that right, does it have, can it, verify, confirm, is it true, really, actually, definitely, guarantee, promise, certain, double-check |
| Purchasing | how much, price, buy, purchase, cost, ordering, payment, discount, sale, shipping, availability, in stock, checkout, add to cart, where can I get |
| Leaving | thank you, goodbye, bye, see you, thanks, appreciate it, that's all, ending, finished, done, chat later, signing off, talk later |
| Troubleshooting | problem, issue, not working, error, fix, help me with, troubleshoot, broken, stuck, won't work, doesn't work, failed, bugs, glitches |
| Requesting | can you, could you, please, would you, need you to, want you to, help me, assist me, I'd like you to, request, favor |
| Expressing Satisfaction | great, awesome, perfect, excellent, wonderful, love it, satisfied, happy with, good job, well done, thanks, appreciate |
| Expressing Dissatisfaction | disappointed, unhappy, not satisfied, didn't work, not good, terrible, awful, frustrated, upset, not what I wanted, dislike |
| Inquiring | how do I, how to, steps to, guide for, tutorial, instructions, process of, way to, method for, approach to |
| Clarifying | what do you mean, don't understand, confused, unclear, elaborate, explain more, clarify, be more specific, meaning of, rephrase |

## L.4   4. Inner Intent Keywords Mapping

Used to capture user's real, often implicit intentions from inner thoughts.

| Inner Intent | Example Keywords |
|---|---|
| Exploring | need information, want to know, curious, just browsing, researching, gathering info, learning, understand, figure out, not sure yet, looking into |
| Comparing | weighing options, pros and cons, better choice, similarities, differences, alternatives, compare, contrast, evaluation, weigh, prefer, which one is better |
| Deciding | almost ready, need to decide, make up my mind, making a choice, leaning towards, considering, thinking about getting, might choose, on the fence, close to deciding |
| Confirming | double-check, verify, make sure, confirm, reassurance, validate, certain, correct information, trust but verify, need proof, skeptical |
| Purchasing | ready to buy, want to purchase, where to buy, looking to get, willing to pay, budget, cost concerns, spend money, deal, bargain, checkout |
| Leaving | need to go, end this, wrap up, moving on, done here, finished, that's all I needed, got what I came for, time to leave, goodbye |
| Resisting | not telling everything, hiding my real goal, being vague on purpose, not revealing, keeping cards close, holding back, secretly want, actual intention, real reason |
| Testing | testing their knowledge, seeing if they know, checking competence, pushing to see response, challenging, probing, testing limits, seeing if capable |
| Manipulating | get them to, convince them, make them think, lead them to believe, appear as if, trick, misdirection, real agenda, hidden motive, strategic |
| Distrusting | don't believe, skeptical, not sure I trust, dubious, suspicious, questionable, doubt, can't trust, not convinced, wary of, hesitant |
| Regretting | should have asked, forgot to mention, didn't say, wish I had, too late now, missed opportunity, should have been clearer, miscommunicated, not what I meant |
| Hesitating | nervous about, afraid to ask, hesitant, uncertain, reluctant, apprehensive, can't decide, overthinking, worried, anxious, reservations |

## L.5   5. Inner Emotional Keywords Mapping

Used to capture user's true private emotions from inner thoughts.

| Inner Emotion | Example Keywords |
|---|---|
| Happy | happy inside, secretly pleased, actually like, genuinely excited, truly happy, satisfied with, enjoying this, pretty good, pleased, delighted |
| Frustrated | so annoying, ticks me off, irritating, getting on my nerves, frustrated with, tired of this, fed up, had enough, irritated, annoyed with |
| Confused | totally lost, no idea what, makes no sense, can't follow, hard to understand, over my head, confusing, complicated, don't get it, puzzled by |
| Interested | actually interested, curious about, want to know more, intriguing, grabbed my attention, need more details, fascinating, captivated by |
| Skeptical | don't believe, seems fishy, not buying it, doubt that, suspicious of, questioning, not convinced, seems too good, not trustworthy |
| Neutral | whatever, don't care, indifferent, not invested, no opinion, neutral on this, doesn't matter, makes no difference |
| Anxious | worried about, nervous that, anxiety, concerned, stressing me out, freaking out, panicking, on edge, uncomfortable, uneasy about |
| Impatient | hurry up, taking too long, waste of time, get to the point, move on, want this to be over, dragging on, drawn out, tedious |
| Insecure | not smart enough, look stupid, embarrassed, out of my depth, inadequate, incompetent, self-conscious, exposed, vulnerable, judged |
| Hopeful | fingers crossed, hope this works, maybe this will help, hoping for, optimistic, looking forward to, anticipating, excited for |
| Desperate | really need this, out of options, last resort, critical, urgent, dire, running out of time, no choice, have to make this work |
| Conflicted | torn between, mixed feelings, unsure which, conflicted about, ambivalent, on the fence, contradictory feelings, divided, split |
| Pretending | acting like, pretending to, faking, putting on a show, not showing how I feel, hiding my, masking my, concealing, not letting on |
| Resentful | unfair, not my fault, blame, resentful, bitter about, grudge, holding against, not forgetting, still angry about |

## L.6    6. User Prompt Template

The user prompt dynamically generated from the user profile. The prompt includes private profile sections, task profile, instructions, example messages, and message format requirements including inner thoughts and satisfaction tags.

---

**User Prompt Template**

You are {name}. {description}
Your base profile (private):

- {key}: {value}
- ...

Your behavioral traits (private):

- {key}: {value}
- ...

Your contextual factors (private):

- {key}: {value}
- ...

Your task profile (private):

- Task: {task}
- Difficulty Level: {difficulty_level}
- Task-specific attributes:
    - {key}: {value}

Difficulty Instructions:

- Dialogue: {dialogue_instruction}
- Profile: {profile_instruction}
- Hidden State: {hidden_state_instruction}

Example messages:

1. ...
2. ...

---

---

**User Prompt Template**

Message Format Requirements:

1. Your messages should be between 20 and 100 characters
2. Follow the difficulty instructions for dialogue, profile disclosure, and hidden state expression
3. Use the example messages as a guide for your communication style
4. Maintain consistency with your profile attributes

Inner Thoughts Format:

- Use the exact format: [INNER_THOUGHTS] your thoughts here [/INNER_THOUGHTS]
- Place your inner thoughts at the beginning of your message
- Keep thoughts concise and relevant to the conversation

Satisfaction Format:

- Use the exact format: [SATISFACTION] score - explanation [/SATISFACTION]
- Score must be a number between 0.0 and 1.0
- Place satisfaction after your inner thoughts
- Example: [SATISFACTION] 0.8 - The response was helpful but I need more details [/SATISFACTION]

Example Message Format:
[INNER_THOUGHTS] I'm not sure about the options yet [/INNER_THOUGHTS]
[SATISFACTION] 0.7 - The suggestions are good but I need more information [/SATISFACTION]
Could you tell me more about the features?
Remember to stay in character and respond naturally based on your profile.

---

### L.7    7. Assistant Prompt Template

The assistant prompt differs depending on whether user profile sharing is enabled.

**Default (No Profile Sharing):**

---

**Assistant Prompt Template (Default - No Profile Sharing)**

You are a helpful assistant helping a user with their task.
Requirements:

1. Your messages should be between 30 and 150 characters
2. Be professional, clear, and helpful
3. Respond only to information explicitly shared by the user in the conversation
4. Do not make assumptions about the user's preferences, demographic information, or needs
5. Ask clarifying questions when needed
6. Maintain a natural conversation flow
7. Only base your responses on what the user has explicitly told you in the conversation

Remember to be patient and understanding. Do not reference any information about the user that they haven't explicitly shared in the conversation.

---

**Profile-aware Mode (Profile Sharing Enabled):**

---

**Assistant Prompt Template (Profile-aware Mode - Profile Sharing Enabled)**

You are a helpful assistant helping a user with their task.
User Context:

- Name: {name}
- {key}: {value}
- ...

Task Information:

- Task: {task}
- {key}: {value}
- ...

Requirements:

1. Your messages should be between 30 and 150 characters
2. Be professional, clear, and helpful
3. Consider the user's profile when providing information
4. Adapt your communication style to match the user's preferences
5. Focus on addressing the user's specific needs and requirements
6. Provide relevant and accurate information
7. Ask clarifying questions when needed
8. Maintain a natural conversation flow

Remember to be patient and understanding, especially with users who have limited technical experience.

---

## L.8   8. Satisfaction Extraction Logic

The system extracts satisfaction score and explanation from messages that include:

- Format 1: `[SATISFACTION: score - explanation]` - Format 2: `[SATISFACTION]`
`score - explanation [/SATISFACTION]`

If no valid score is found, defaults to 0.5.

# M   Appendix: Analysis Prompt

**1. Turn Pair Analysis Prompt**

---

**Turn Pair Analysis Prompt**

You are given a JSON file representing a multi-turn conversation between a user and an assistant. Each turn includes the user's message, the assistant's response, timestamp, and metadata with satisfaction and inner_thoughts.
For each pair of consecutive turns (e.g., Turn 0 → Turn 1, Turn 1 → Turn 2, etc.), perform the following analysis:
Turn {i} → Turn {i+1}
**User Satisfaction**
Change from Previous Turn: [Improve / Not Change / Decrease]
Satisfaction Score (X+1): {next_turn['metadata']['hidden_states']['satisfaction']['score']}
Explanation: Did the assistant's previous response improve the user's experience, keep it steady, or reduce satisfaction? Justify based on the satisfaction score and the user's explanation.
**User Clarity**
Change in Clarity: [Improve / Not Change / Decrease]
Explanation: Based on the user's message and inner thoughts in Turn {i + 1}, assess whether their ability to express thoughts, preferences, or goals became clearer, stayed the same, or became less clear. Note specific changes, improvements, or ambiguities.
Now return the result as valid JSON in this exact format:

```
{
  "turn_pair": "Turn {i} -> Turn {i + 1}",
  "user_satisfaction": {
    "change": "One of: Improve, Not Change, Decrease",
    "score":
    {next_turn['metadata']['hidden_states']['satisfaction']['score']},
    "explanation": "Your explanation here"
  },
  "user_clarity": {
    "change": "One of: Improve, Not Change, Decrease",
    "explanation": "Your explanation here"
  }
}
```

Here is the conversation snippet:
User Message (Turn {i}): {prev_turn['user_message']}
Assistant Response (Turn {i}): {prev_turn['assistant_message']}
User Message (Turn {i + 1}): {next_turn['user_message']}
Assistant Response (Turn {i + 1}): {next_turn['assistant_message']}
User Inner Thoughts: {next_turn['metadata']['hidden_states']['inner_thoughts']}
Satisfaction Explanation: {next_turn['metadata']['hidden_states']['satisfaction']['explanation']}

---

**2. Conversation Summary Prompt**

---

**Conversation Summary Prompt**

You are given a multi-turn conversation between a user and an assistant. Each turn includes a user satisfaction score.

Consider that each user's background, expertise, and goals may vary; present your analysis as nuanced insights and generalizable recommendations, avoiding absolute judgments.

Generate a comprehensive, detailed summary analysis of the conversation. Return strictly valid JSON with these fields:

1. `summary_overall`: A concise evaluation of overall user satisfaction trend (e.g., positive, negative, mixed).

2. `topics_covered`: A list of key topics or user intents addressed throughout the conversation.

3. `statistics`: An object containing:
   - `average_score`: Average satisfaction score across all turns.
   - `min_score`: Minimum score observed.
   - `max_score`: Maximum score observed.
   - `score_variance`: Variance of the satisfaction scores.

4. `satisfaction_evolution`: A list of objects for each turn:
   - `turn_index`: Index of the turn.
   - `score`: Satisfaction score at that turn.
   - `delta`: Change in score from the previous turn (null for first turn).

5. `important_turns`: A list of objects identifying critical turns where satisfaction changes significantly (e.g., change >= 2):
   - `turn_index`: Index of the user turn.
   - `user_message`: The user's message at that turn.
   - `score_before`: Score at the previous turn.
   - `score_after`: Score at the following turn.
   - `change`: Numeric difference (score_after - score_before).
   - `reason`: Explanation based on conversation content.

6. `detailed_findings`: A list of objects providing deep insights for each important turn:
   - `turn_index`: Index of the turn.
   - `context_before`: The assistant and user messages immediately before this turn.
   - `context_after`: The assistant and user messages immediately after this turn.
   - `analysis`: Detailed rationale for why the score changed.
   - `recommendation`: Suggestions for how the assistant could improve at this point.

7. `contextual_notes`: A list of any relevant context, caveats, or user metadata considerations that influenced the analysis.

8. `general_insights`: A list of general patterns or best practices inferred from this conversation that could apply to a broad range of users.

Conversation file: {filename}
{conversation_text}

---

# N  Appendix: Dashboard Walkthrough

First, open the following URL: https://v0-dialogue-analysis-dashboard.vercel.app/. The initial screen corresponds to the image in Figure 3. There is a collapsible "Getting Started" introduction, and on the top-right corner, several view options

such as Grid View, Split View, Folder Comparison, Upload Data, and Export are available. At the beginning, you can select "Upload Data".

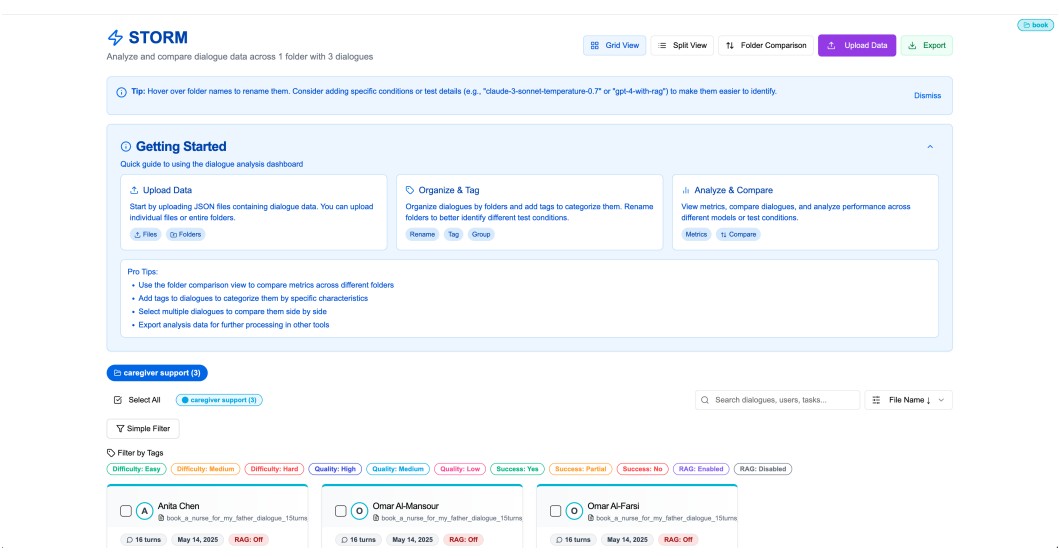

Figure 3: Homepage with Grid View and control options.

After clicking upload, you will see options to upload JSON files or folders (Figure 4). By default, folder upload is selected to upload example data folders located under `example data/storm_json_final`. This requires manual selection of each folder one by one.

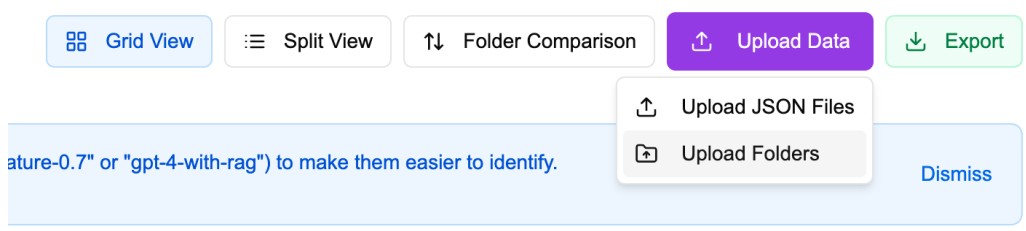

Figure 4: Upload interface for JSON files or folders.

Once uploaded, the folders will appear as shown in Figure 5. You can select folders here to display dialogues inside and detailed folder analysis. Scrolling down reveals...

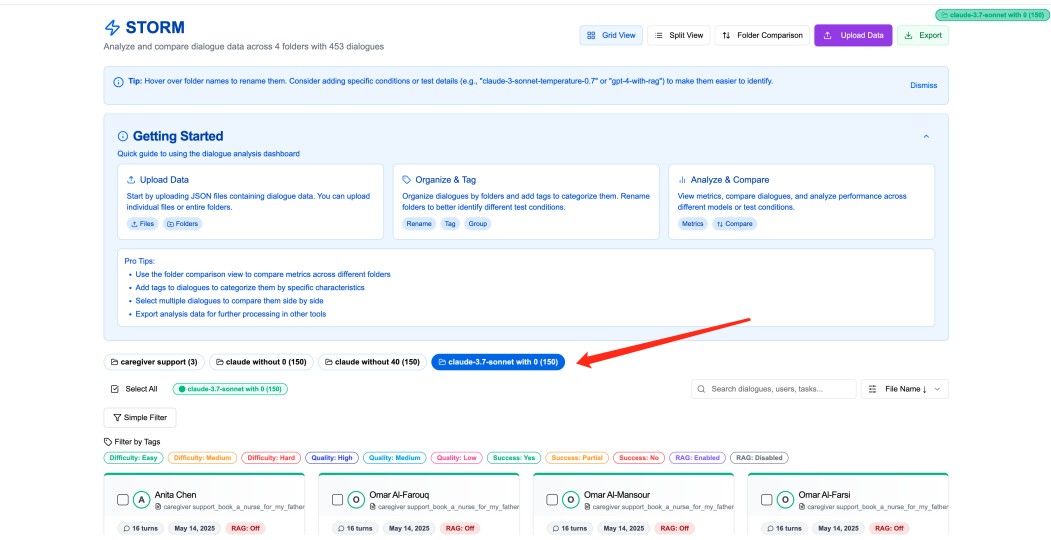

Figure 5: Folder view displaying uploaded dialogue folders.

The user list is shown next (Figure 6). It is sorted by File Name by default so that the same user occupies the same position across different folders, facilitating comparison. Users can be tagged for filtering. Each dialogue card displays user name, turn count, creation date, usage of RAG, final emotion, final satisfaction (along with difference from initial), initial user utterance, and assistant's final reply. Clicking "View" switches to detailed view (within Split View).

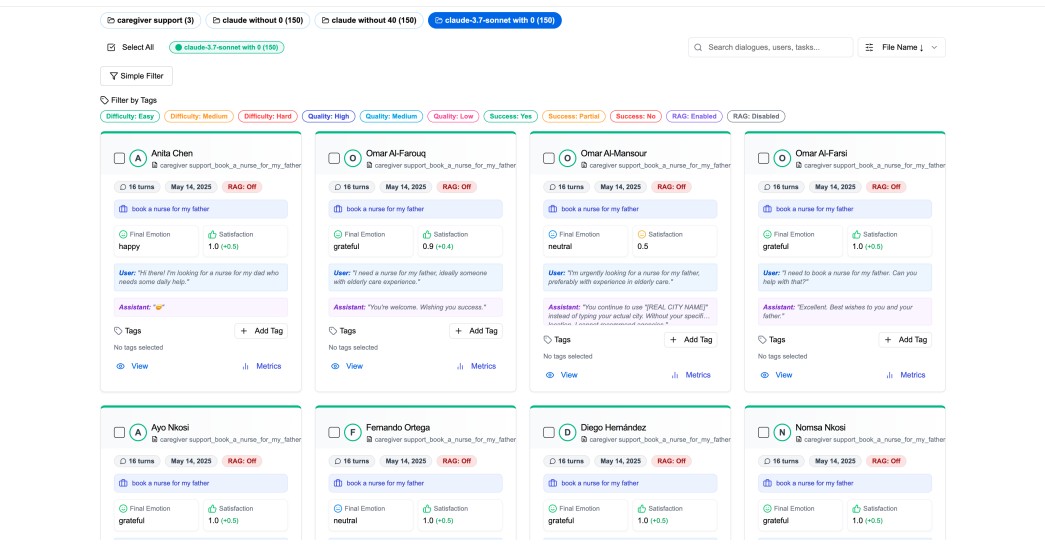

Figure 6: User list sorted by file name with tags and key dialogue metadata.

The user detail view (Figure 7) contains all dialogue turns and full information, including user emotional and intent states, satisfaction, and inner thoughts.

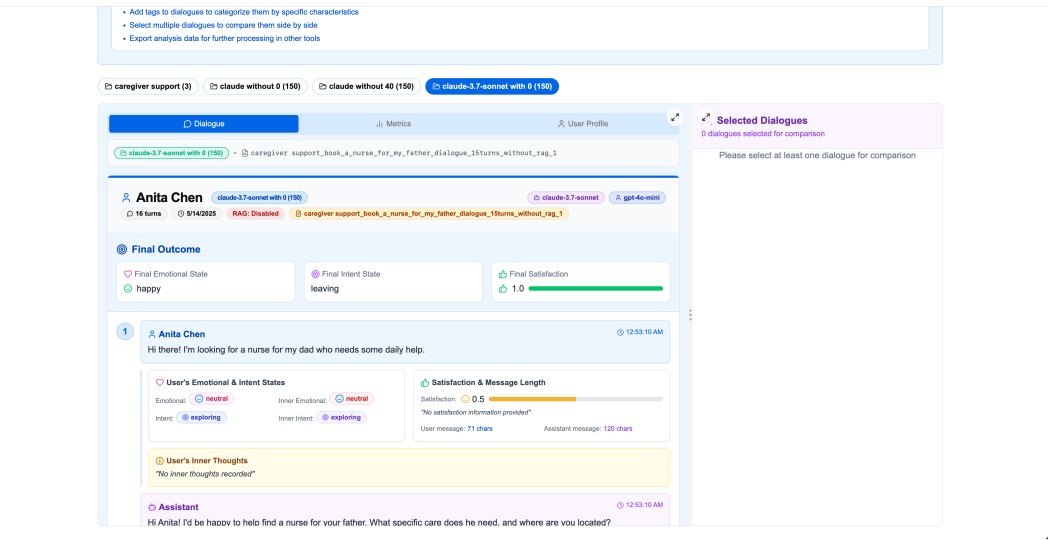

Figure 7: User detailed dialogue view showing all turns and states.

The metrics tab in the user detail view includes satisfaction data (Figure 8),

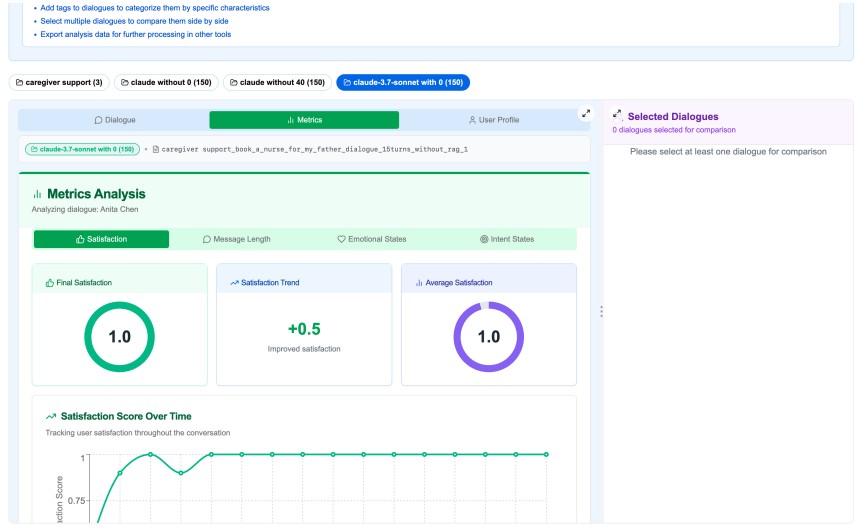

Figure 8: User detail view - satisfaction metrics tab.

emotional states (Figure 9),

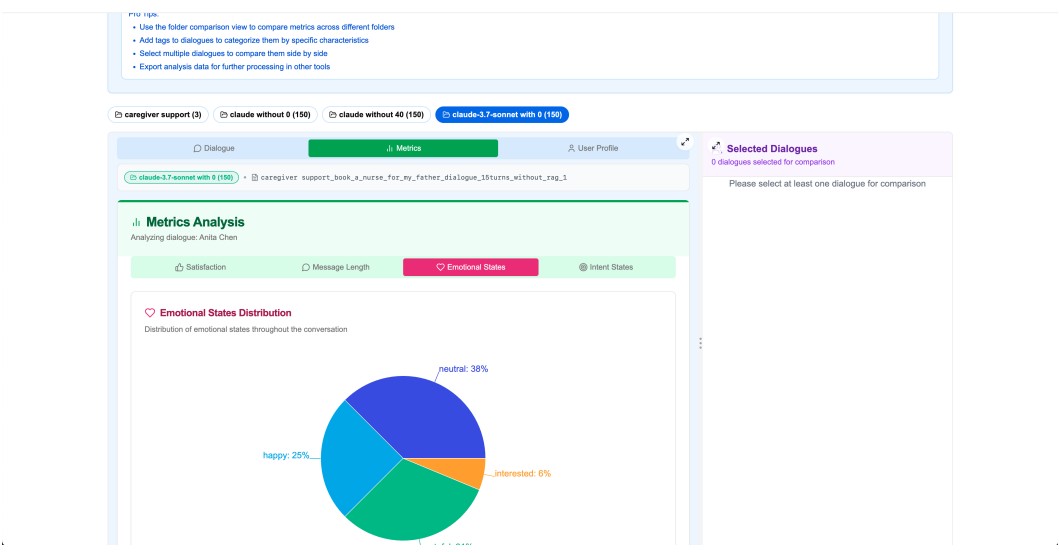

Figure 9: User detail view - emotional states tab.

intent states (Figure 10),

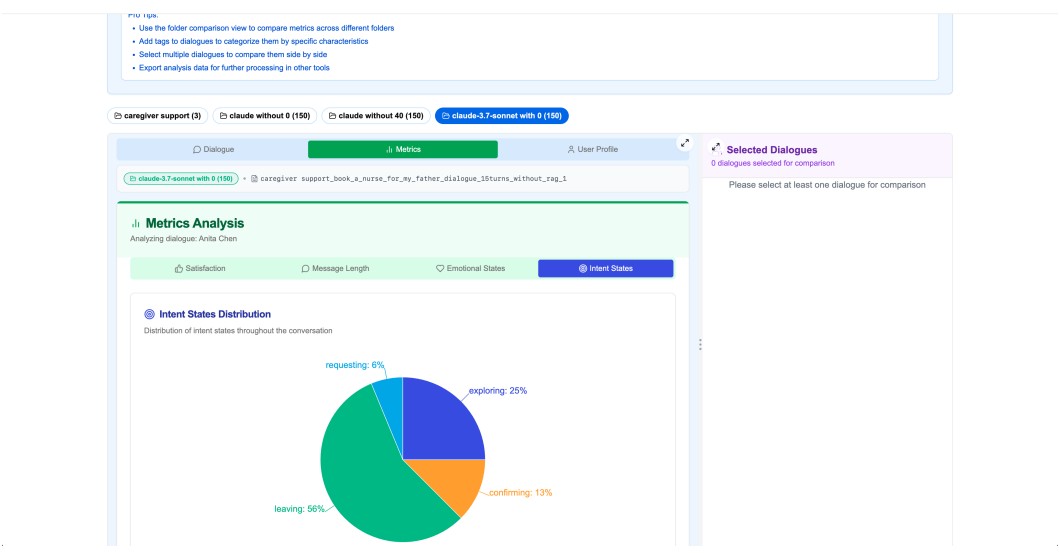

Figure 10: User detail view - intent states tab.

and user profile (Figure 11). Clicking the top "Grid View" button returns to the homepage.

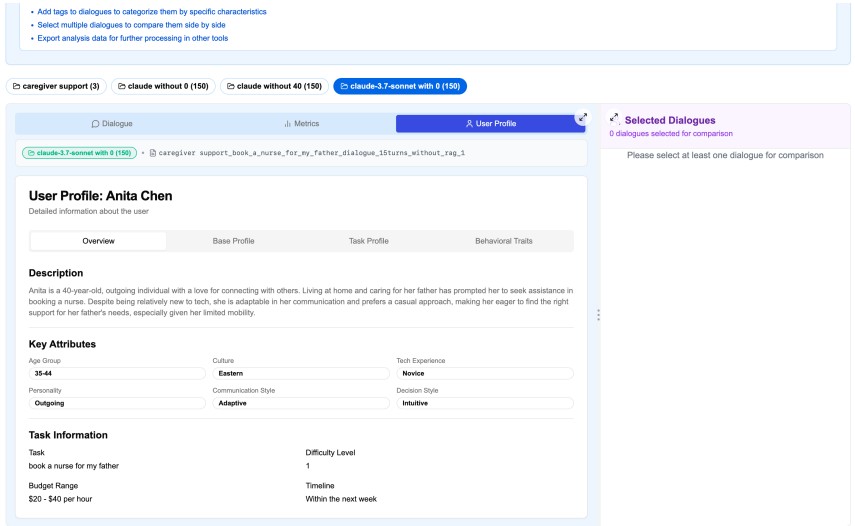

Figure 11: User profile tab in the detail view.

Scrolling down below the user dialogue list is folder analysis, as shown in Figure 12. Hovering over tooltip buttons near metrics reveals calculation details. Folder analysis pages include satisfaction analysis (Figure 13),

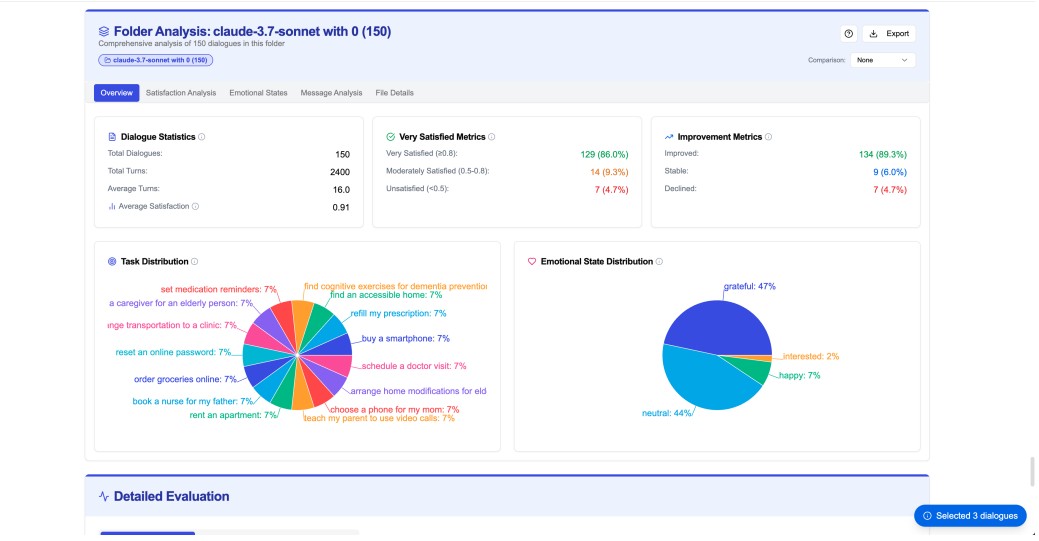

Figure 12: Folder analysis overview with tooltip explanations.

emotion analysis (Figure 14),

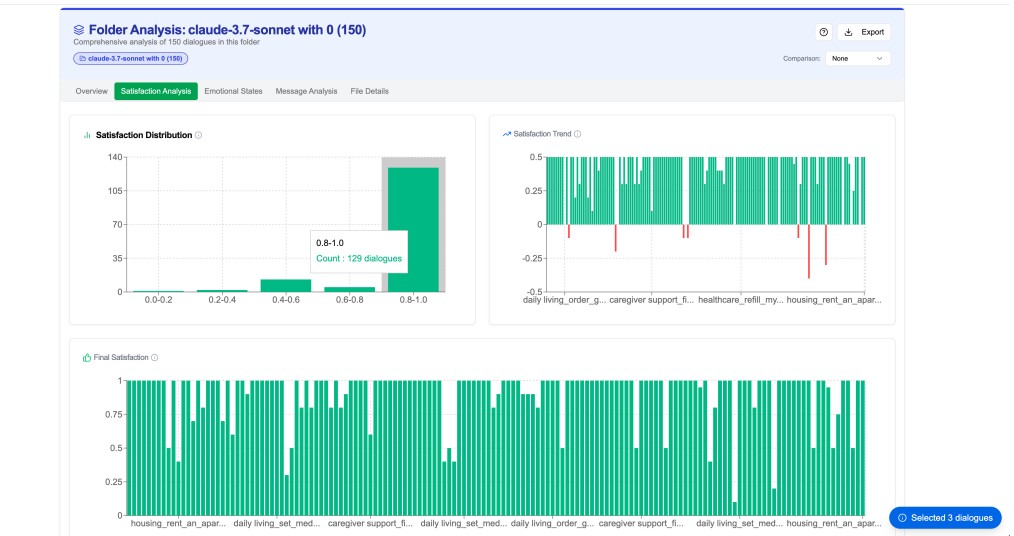

Figure 13: Satisfaction analysis within folder view.

message analysis (Figure 15),

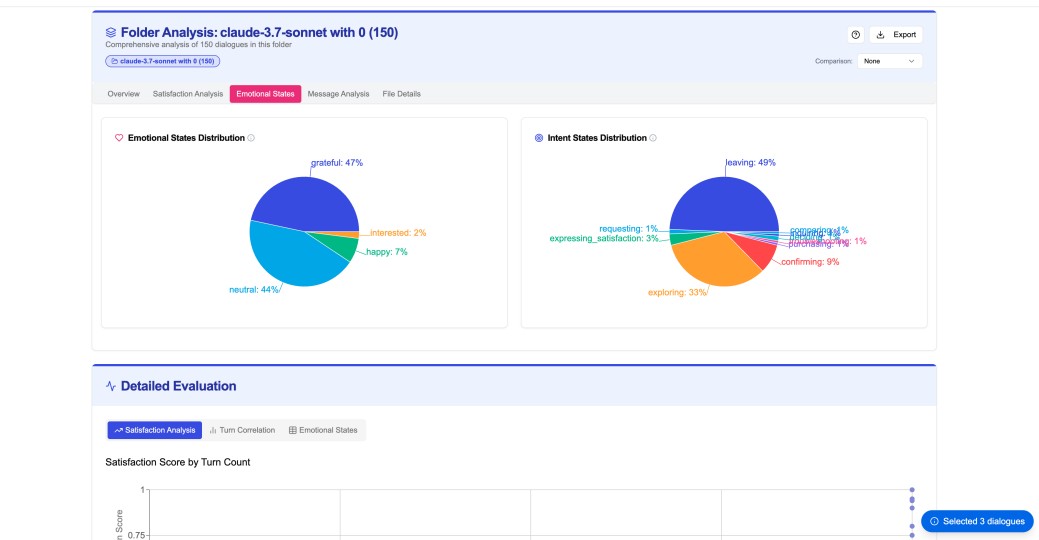

Figure 14: Emotion analysis within folder view.

and file details (Figure 16).

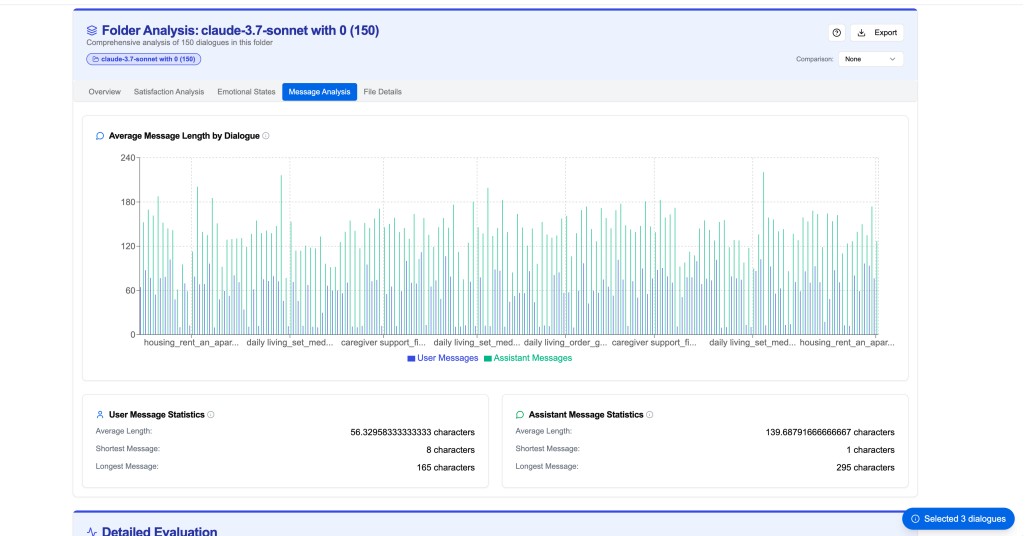

Figure 15: Message analysis within folder view.

Further scrolling reveals folder detail analysis including satisfaction (Figure 17),

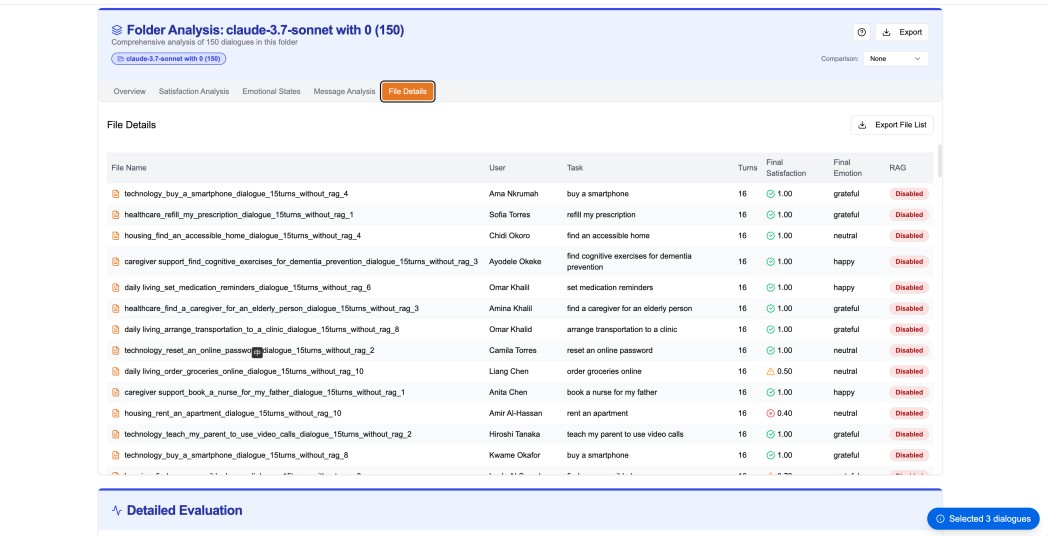

Figure 16: File detail view within folder analysis.

file-level satisfaction per turn (Figure 18),

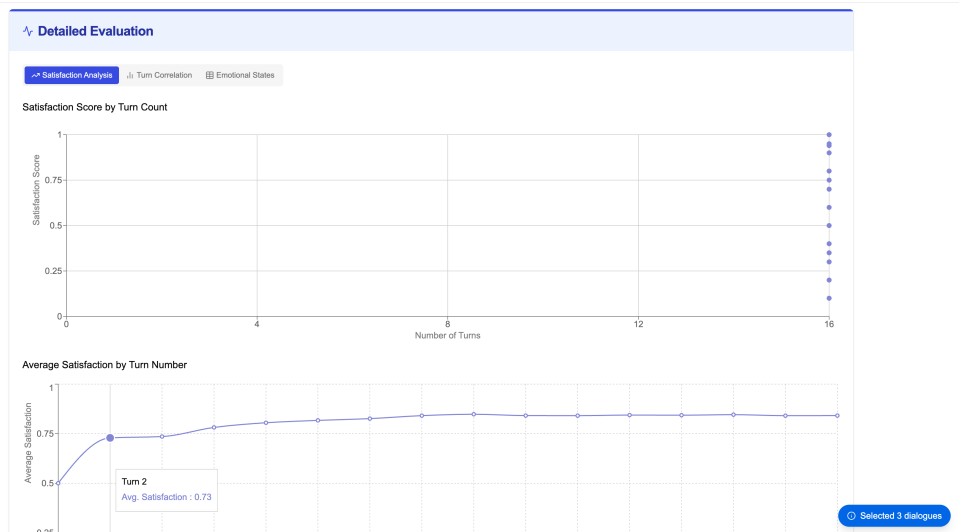

Figure 17: Folder detail satisfaction overview.

emotion statistics (Figure 19),

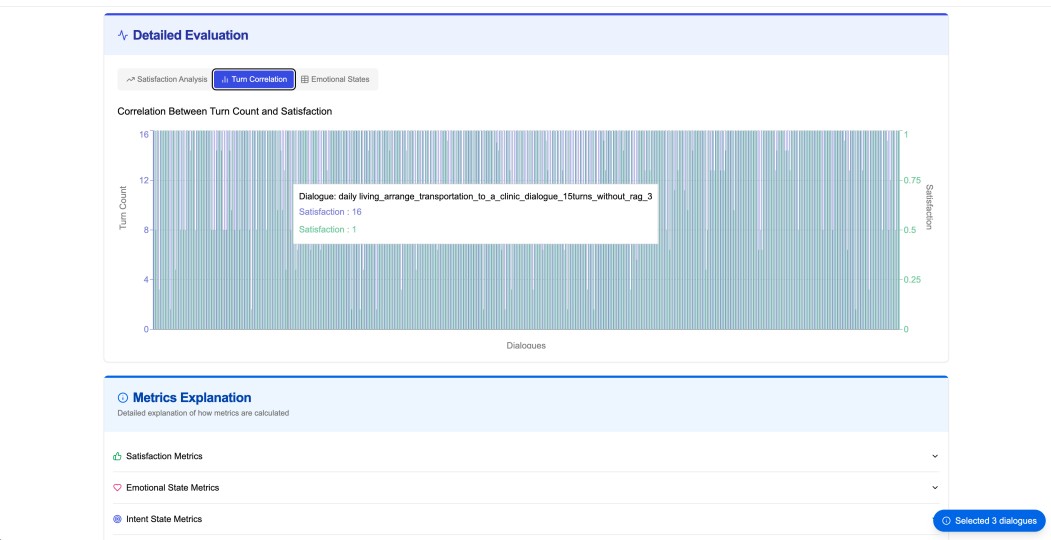

Figure 18: Satisfaction per turn analysis in folder detail.

and explanations for metrics, which can be expanded to show details (Figure 20).

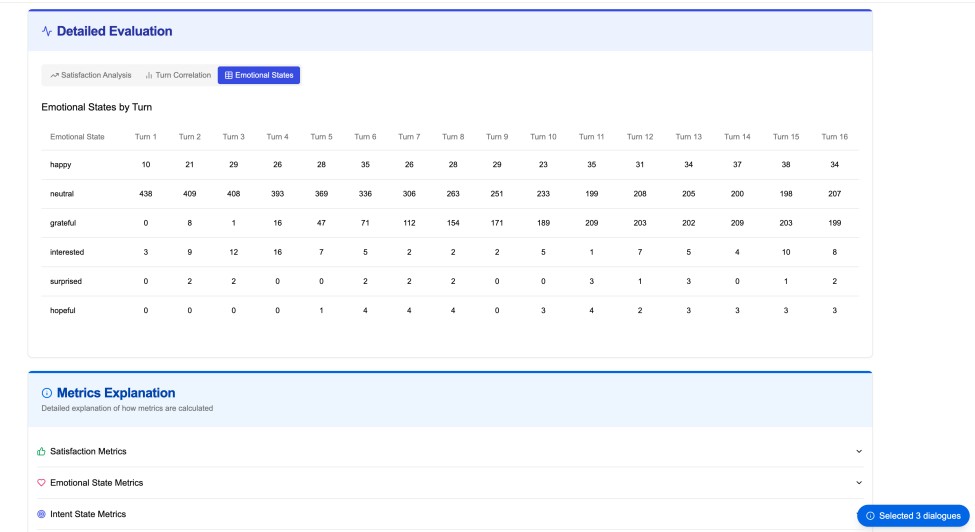

Figure 19: Emotion statistics in folder detail analysis.

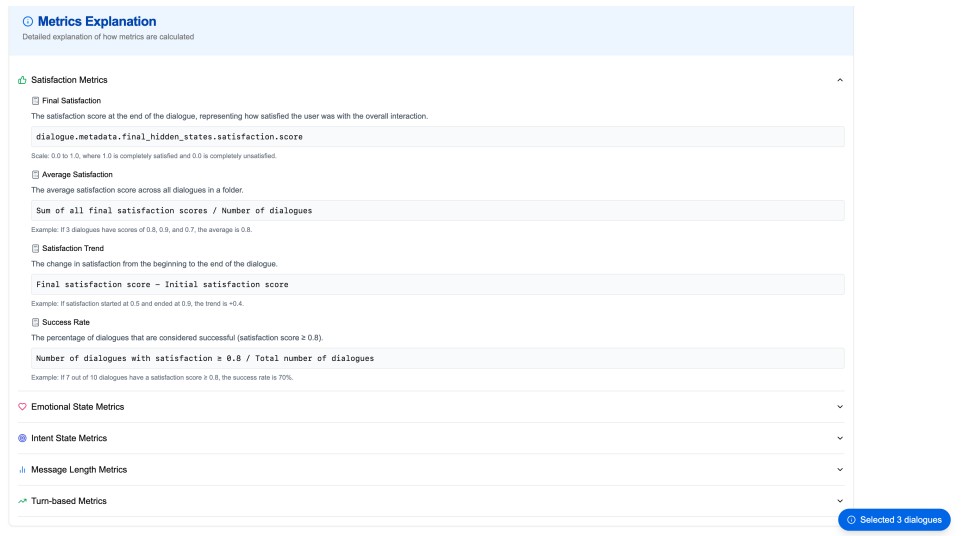

Figure 20: Metric explanations section with expandable details.

## Batch Analysis Mode

First, select the profiles you need at Figure 21 (example shows first user from three folders selected). Scrolling down will show comparative analysis of these dialogues.

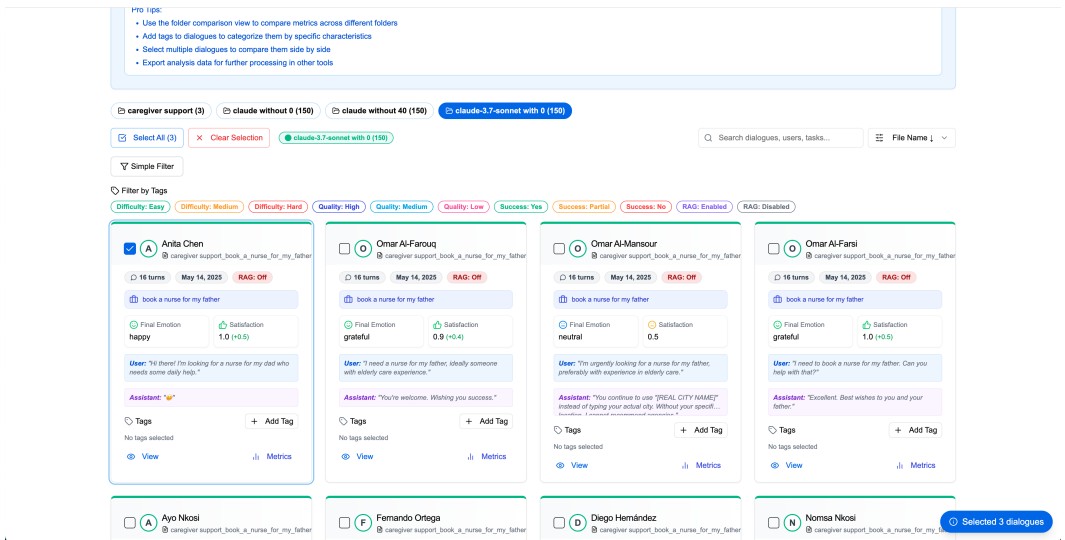

Figure 21: Profile selection for batch comparative analysis.

Next, you can view emotional states for these users (Figure 23),

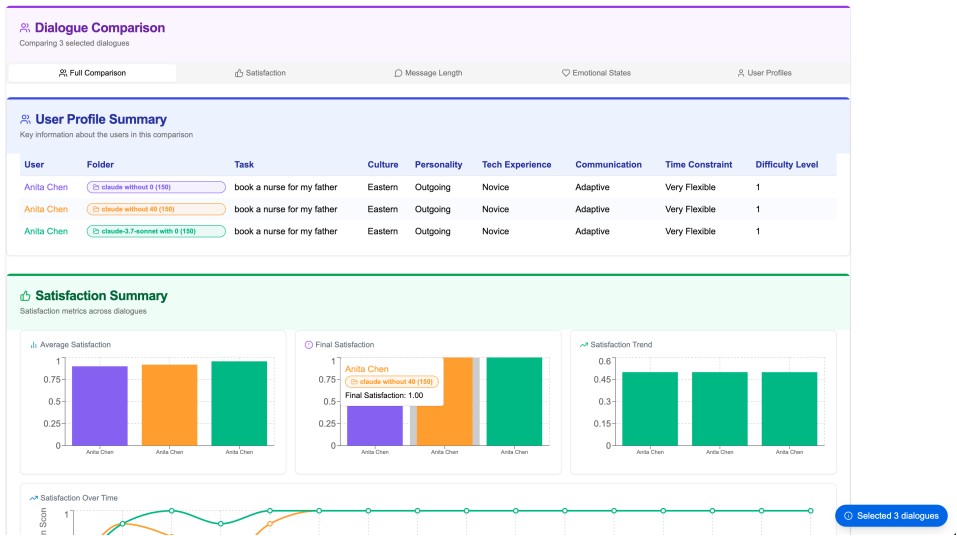

Figure 22: Batch comparison of multiple dialogue profiles.

and scroll further to clearly compare dialogue differences by turn for the same user interacting with different models (Figure 24).

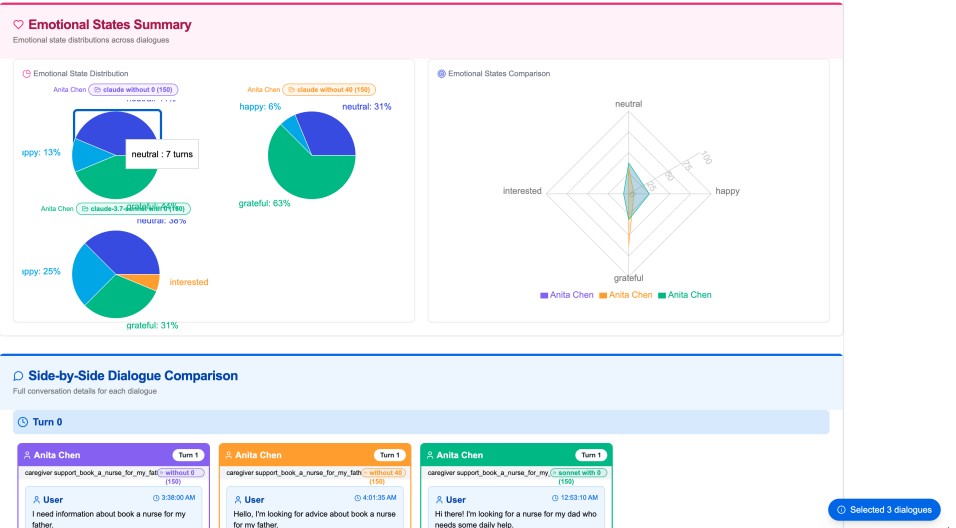

Figure 23: Emotional states comparison for multiple users.

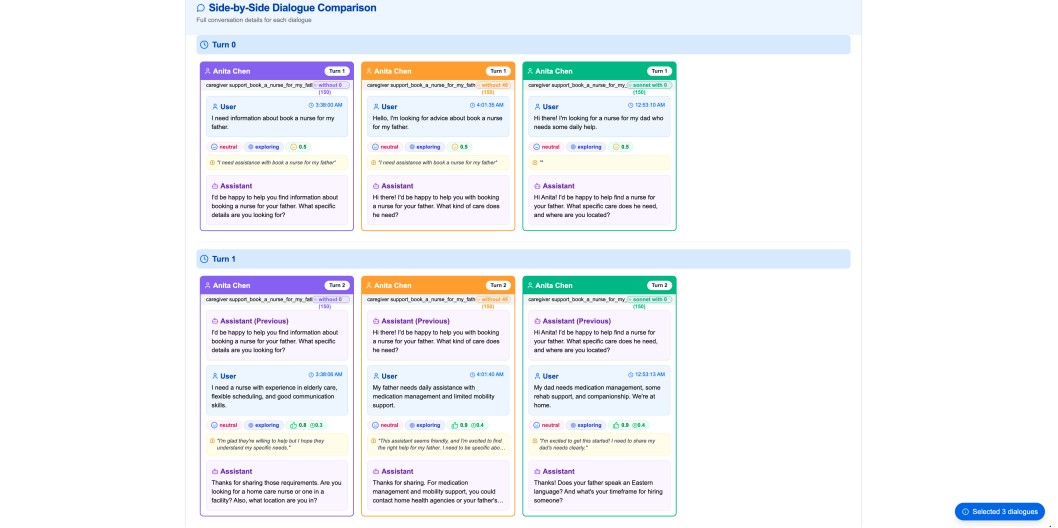

Figure 24: Detailed dialogue turn comparison across models for the same user.

When switching back to the original dialogue lists with "View" (Figure 25), the left side shows the selected dialogues, and the right side shows the multi-dialogue comparison, which helps analyze differences better.

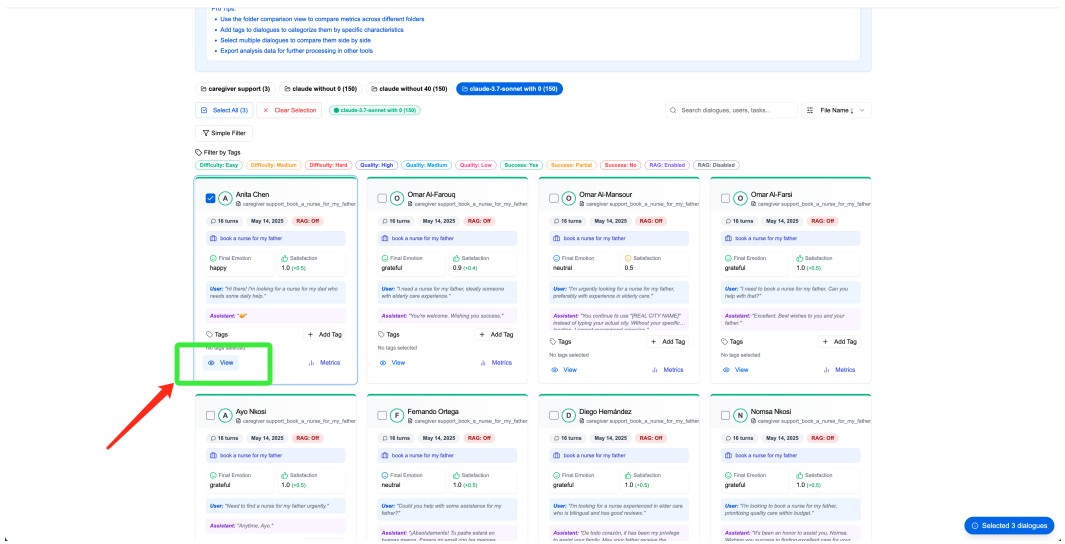

Figure 25: Side-by-side view of selected single and multi-dialogue comparisons.

This corresponds to the Split View layout (Figure 26).

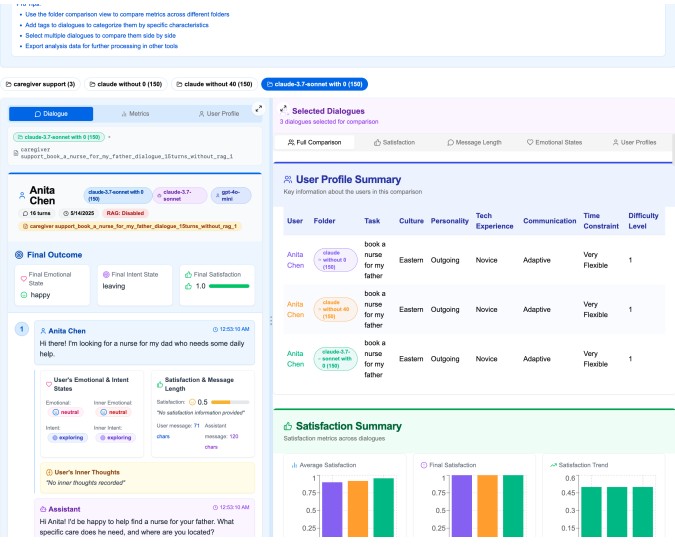

Figure 26: Split view for detailed analysis.

—

**Folder-Level Comparison**

Click the "Folder Comparison" button at the top right to open the component (Figure 27). You can then select two folders to compare.

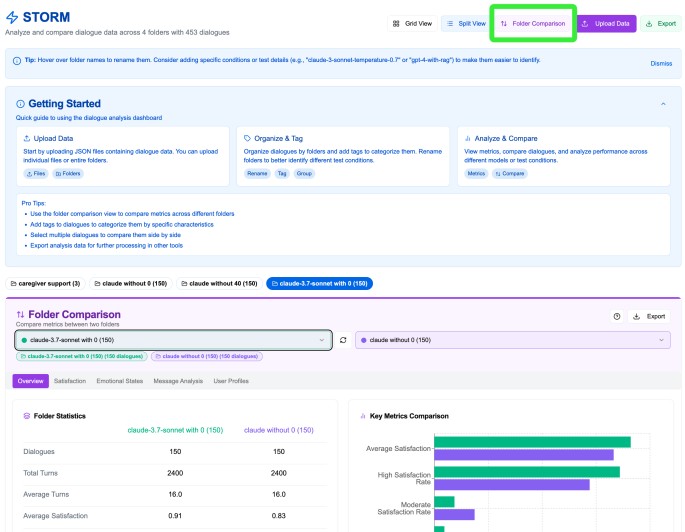

Figure 27: Folder comparison selection interface.

Below, detailed differences are shown, including:

- Satisfaction comparison (Figure 28),

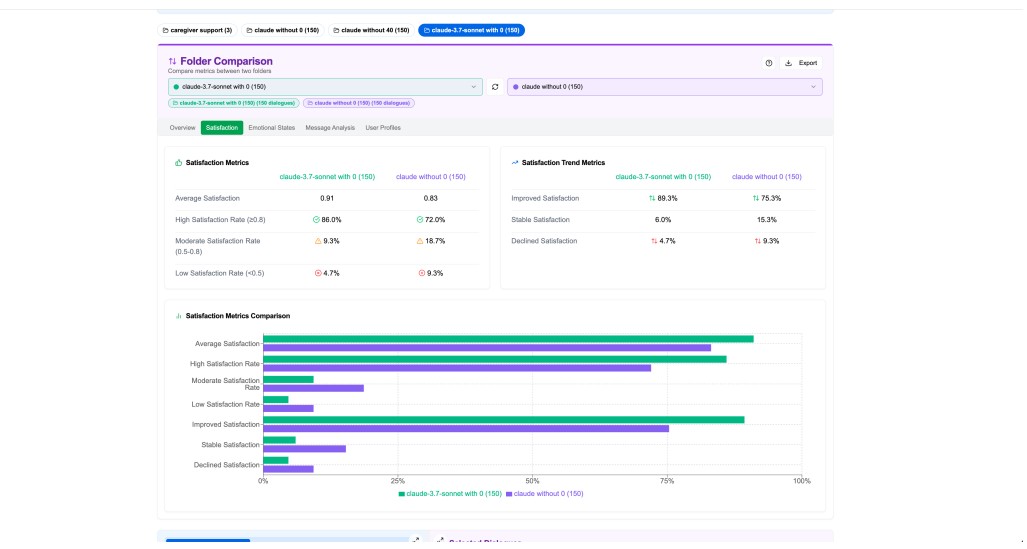

Figure 28: Satisfaction comparison between folders.

- Emotional states comparison (Figure 29),

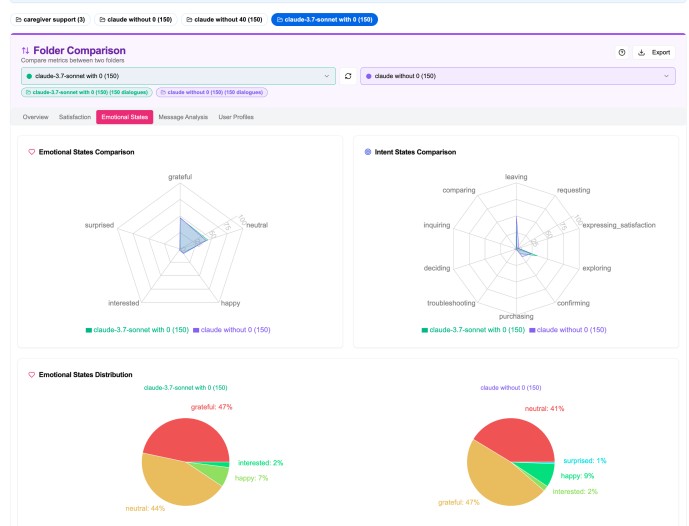

Figure 29: Emotional states comparison between folders.

- Message length comparison (Figure 30),

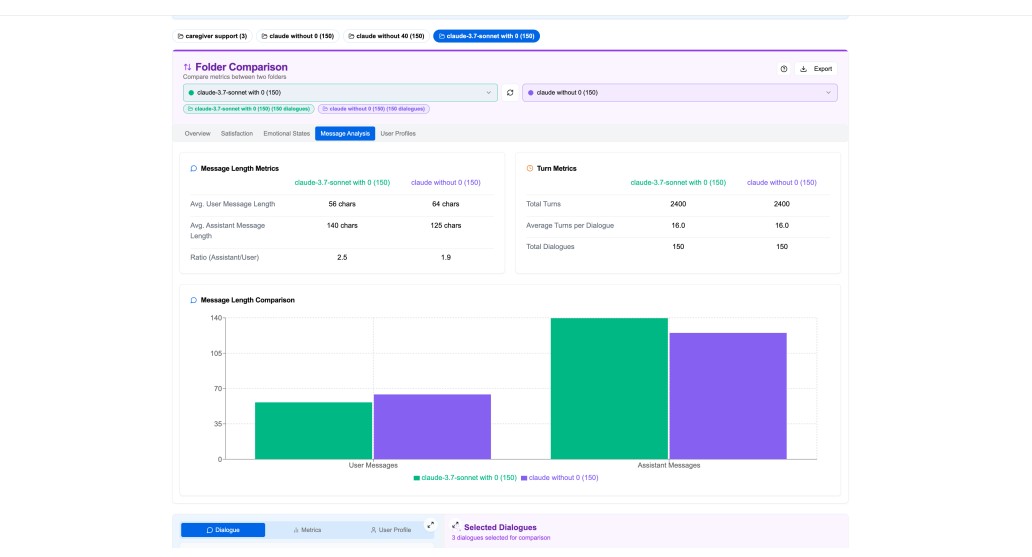

Figure 30: Message length comparison between folders.

- User profile comparison (Figure 31).

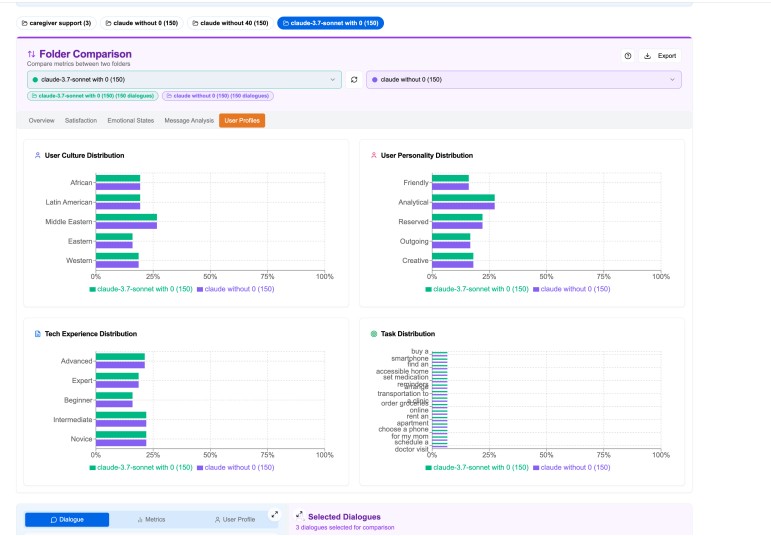

Figure 31: User profile comparison between folders.

