# OpenReview forum: "WHEN TO ACT, WHEN TO WAIT: Modeling the Intent-Action Alignment Problem in Dialogue"
_colmweb.org/COLM/2025/Workshop/Social_Sim — Social Sim'25_

### Official Review · Reviewer_szmn · 2025-07-04

**Rating:** 3
**Overall Assessment:** 1
**Confidence:** 4

**Review:**

1. **Excessive Use of Undefined, Jargon-like Terminology**
   The paper is saturated with ambiguous terms that resemble conceptual padding rather than grounded technical constructs. This issue is pervasive enough to obstruct the reader’s ability to discern what the authors are actually claiming or measuring. Many of these terms are presented as self-evident but are not defined, cited, or operationalized in any replicable way. The result is a loss of scientific clarity and a failure to meet basic standards for rigor and falsifiability. Examples include:

   - **"Structural information" / "structural signals"** – Unclear what level of structure is intended (syntax, discourse, task logic, etc.)
   - **"Contextually triggerable expression"** – No explanation of what constitutes triggerability, nor what contextual conditions apply
   - **"Expression trajectories"** – Undefined; unclear whether this refers to semantic drift, intent formulation, or surface phrasing
   - **"Latent cognitive transitions"** – Vague and unmeasurable; no explanation of what transitions are occurring or how they’re inferred
   - **"Asymmetric information processing"** – Introduced without definition in the abstract and intro
   - **"Internal cognitive improvements"** – Not defined or grounded in psychological or interactional literature
   - **"Optimal information completeness"** – Vague optimization target with no theoretical or empirical framing
   - **"Asymmetric reasoning dynamics"** – Not specified; appears to restate earlier "asymmetric information" with added vagueness
   - **"Stability intervals"** – Not explained; if meant as a technical term, needs definition and grounding
   - **"Expressions"** – Used ambiguously; unclear whether interchangeable with "utterances" or meant to include unspoken user state

2. The statement that “current systems cannot access embedded contextual cues” (L36) misrepresents the literature. Dialogue state tracking, contextual NLU, user simulation, and multi-turn reasoning are established research areas addressing precisely these issues. The paper’s sweeping dismissal is not only unsubstantiated but ignores decades of work in task-oriented dialogue systems and discourse-aware modeling.
3. The claim that existing systems treat intent as "binary" (L31) is misleading. Intent classification may use discrete labels, but this does not imply that models treat intent as binary in a conceptual or functional sense. No citation is given to support this characterization, and it contradicts the diversity of techniques used in NLU, such as multi-label classification, hierarchical intent modeling, and probabilistic intent ranking.
4. The entire evaluation pipeline is synthetic, relying on simulated “inner thoughts” and profile-driven generation. No human annotations, real users, or externally verifiable ground truth are included. This introduces circularity: the same model defines the user’s intent, evaluates the agent’s response, and declares improvement.
5. The paper claims to model intent along a “continuous spectrum” but does not clarify what dimensions this spectrum involves (semantic specificity, task readiness, goal clarity?). Without an explicit formalization, the claim remains rhetorical.

The writing does not meet basic standards for scientific clarity. Many terms are overloaded, inconsistent, or invented without definition. This is not an issue of English fluency but of conceptual rigor. Without clearer exposition and proper citation of theoretical constructs, the work risks confusing readers and misrepresenting the field. Substantial rewriting is necessary before the paper is suitable for publication.

**Comments Suggestions And Typos:**

1. Cite Subramonyam right after introducing the "gulf of envisioning" phrase in L24
2. L29: "Subramonyam et al. Subramonyam et al. (2023)" is redundant

**Paper Summary:**

The paper introduces STORM, a framework for modeling evolving user intent in task-oriented dialogue through simulated interactions between two language models with asymmetric information access. It aims to capture when user expressions become actionable by generating annotated dialogues that include internal user states and clarity metrics. The authors propose new evaluation methods to assess both task performance and cognitive alignment between users and agents. They also examine how varying user profile completeness affects model behavior, suggesting that moderate uncertainty can improve interaction quality.

**Relevance:**

3

**Summary Of Strengths:**

1. Frames intent as a continuous process, addressing limitations of binary intent classification in dialogue systems.
2. Allows configurable simulation of user variability through profile attributes and expression difficulty levels.
3. Offers a reproducible setup for generating synthetic dialogues with annotated internal states, useful for controlled experimentation.

**Summary Of Weaknesses:**

The paper introduces too many loosely connected components, terms, and mechanisms, which collectively obscure rather than clarify its core contributions. The conceptual framing is overloaded with undefined jargon, and the central problem statement is difficult to extract. Critically, the work is almost entirely detached from the existing task-oriented dialogue literature. It neither engages with established research on intent modeling, dialogue state tracking, or clarification strategies, nor justifies how its approach improves upon them. As a result, the paper lacks both theoretical grounding and empirical relevance, leaving its practical value unclear.

---

### Official Review · Reviewer_maHo · 2025-07-04
**Submission24 Official Review**

**Rating:** 7
**Overall Assessment:** 4
**Confidence:** 4

**Review:**

The paper overall is in good shape. It's well written and easy to follow, and the components are explained very well. The pros and cons will be discussed in the next sections.

**Comments Suggestions And Typos:**

N/A

**Paper Summary:**

This paper presents STORM (Structured Task-Oriented Representation Model), aiming to

1. formalize asymmetric information processing in dialogue systems and
1. model the intent formation tracking collaborative understanding evolution.

The motivation is that users sometime make assumptions when they provide tasks to the agents, which most of the time require detailed, fine-grained execution details. The system consists of three components:
1. a dialogue generation pipeline,
1. a database-driven memory system and
1. a web-based dialogue visualization interface.

To model the userLLM, they used GPT-4o Mini, and models the user profile as a composite structure: *(task-agnostic components, task-dependent components, communication modeling components)*.

To model the agentLLM, they incorporate the agent role and generate agent responses. Since the agent does not have access to the user hidden states, this forms an asymmetric dialogue setting.

They observe that Claude maintains consistent satisfaction across varying profile completeness, Gemini demonstrates robust performance under high uncertainty, while Llama achieves superior intent clarification despite satisfaction trade-offs.

**Relevance:**

5

**Summary Of Strengths:**

1. The paper is well written, with clear annotations and experiment setup. The results are reasonable, presenting some analysis from the results too.

1. I like how the userLLMs are modeled. It included the attributes that  I agree matter the most but not too complicated to result in too many dimensions.

1. They provide live websites to visualize the results .

**Summary Of Weaknesses:**

1. The selection of the user profiles are not discussed in details, or some ablation study changing the user profile dimensions.
1. Some examples demonstrating the user tasks and agent response under different settings.
1. The reliability of the metrics are not thoroughly discussed. How are the metrics quantified? How are they evaluated? Can we trust the numbers? Any human evaluation involved?
1. How successful is the userLLM modeling the unclear user intent? Maybe some examples showing that the model successfully simulated the scenario when a task is not clear but got clarified and resulted in a good response.

---

### Meta-Review · Program_Chairs · 2025-07-24

**Recommendation:** Accept

**Metareview:**

Please incorporate the feedback!